# ProxySPEX: Inference-Efficient Interpretability via Sparse Feature Interactions in LLMs

**Landon Butler**[*]
Department of EECS
UC Berkeley
landonb@berkeley.edu

**Abhineet Agarwal**[*]
Department of Statistics
UC Berkeley
aa3797@berkeley.edu

**Justin Singh Kang**[*]
Department of EECS
UC Berkeley
justin_kang@berkeley.edu

**Yigit Efe Erginbas**
Department of EECS
UC Berkeley
erginbas@berkeley.edu

**Bin Yu**
Departments of Statistics and EECS
UC Berkeley
binyu@berkeley.edu

**Kannan Ramchandran**
Department of EECS
UC Berkeley
kannanr@berkeley.edu

## Abstract

Large Language Models (LLMs) have achieved remarkable performance by capturing complex interactions between input features. To identify these interactions, most existing approaches require enumerating all possible combinations of features up to a given order, causing them to scale poorly with the number of inputs $n$. Recently, Kang et al. (2025) proposed SPEX, an information-theoretic approach that uses interaction sparsity to scale to $n \approx 10^3$ features. SPEX greatly improves upon prior methods but requires tens of thousands of model inferences, which can be prohibitive for large models. In this paper, we observe that LLM feature interactions are often *hierarchical*—higher-order interactions are accompanied by their lower-order subsets—which enables more efficient discovery. To exploit this hierarchy, we propose PROXYSPEX, an interaction attribution algorithm that first fits gradient boosted trees to masked LLM outputs and then extracts the important interactions. Experiments across four challenging high-dimensional datasets show that PROXYSPEX more faithfully reconstructs LLM outputs by 20% over marginal attribution approaches while using $10\times$ *fewer inferences* than SPEX. By accounting for interactions, PROXYSPEX efficiently identifies the most influential features, providing a scalable approximation of their Shapley values. Further, we apply PROXYSPEX to two interpretability tasks. *Data attribution*, where we identify interactions among CIFAR-10 training samples that influence test predictions, and *mechanistic interpretability*, where we uncover interactions between attention heads, both within and across layers, on a question-answering task. The PROXYSPEX algorithm is available at https://github.com/mmschlk/shapiq.

## 1 Introduction

Large language models (LLMs) have achieved great success in natural language processing by capturing complex interactions among input features. Modeling interactions is not only crucial for language, but also in domains such as computational biology, drug discovery and healthcare, which require reasoning over high-dimensional data. In high-stakes contexts, responsible decision-making based on model outputs requires interpretability. For example, in healthcare, a physician relying on LLM diagnostic assistance must intelligibly be able to explain their decision to a patient.

Post-hoc feature explanation methods such as SHAP [1] and LIME [2] focus on marginal attributions and do not explicitly capture the effect of interactions. To address this limitation, recent work

---

[*]Equal contribution. Order determined by coin flip.

39th Conference on Neural Information Processing Systems (NeurIPS 2025).

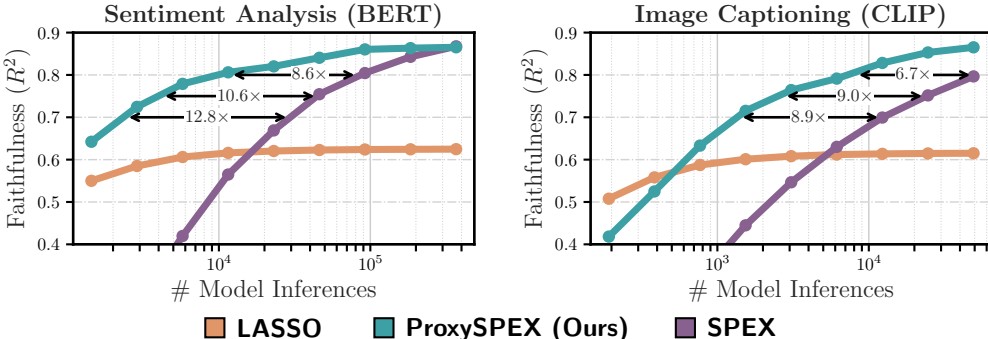

Figure 1: PROXYSPEX requires $\sim 10\times$ fewer inferences to achieve equally faithful explanations as SPEX for a sentiment classification and image-captioning task using a BERT and CLIP model respectively. LASSO faithfulness plateaus indicating limits of marginal approaches.

has proposed interaction indices, such as Faith-Shap [3], that attribute all interactions up to a given order $d$ by exhaustively enumerating them. With $n$ features, enumerating $O(n^d)$ interactions quickly becomes infeasible for even small $n$ and $d$. Kang et al. [4] recently introduced SPEX, the first interaction attribution method capable of scaling up to $n = 1000$ features. SPEX scales with $n$ by observing that LLM outputs are driven by a small number of interactions. It exploits this sparsity by utilizing a sparse Fourier transform to efficiently search for influential interactions without enumeration. For example, with $n = 100$ features, SPEX requires approximately $2 \times 10^4$ model inferences to learn order 5 interactions—a small fraction of all possible $100^5$ interactions. Nonetheless, $2 \times 10^4$ inferences is prohibitively expensive for large models. Hence, the question naturally arises: *Can we identify additional structural properties among interactions to improve inference-efficiency?*

We show empirically that local (i.e., input specific) LLM feature interactions are often *hierarchical*: for an order $d$ interaction, an LLM includes lower-order interactions involving subsets of those $d$ features (see Figure 2). We use this to develop PROXYSPEX, an interaction attribution algorithm that reduces the number of inferences compared to SPEX by $10\times$ while achieving equally faithful explanations. PROXYSPEX exploits this local hierarchical structure by first fitting gradient boosted trees (GBTs) as a proxy model to predict the output of LLMs on masked input sequences. Then, PROXYSPEX extracts important interactions from the fitted GBTs [5].

**Evaluation overview.** We compare PROXYSPEX to marginal feature attributions and SPEX across four high-dimensional datasets with hundreds of features. Results are summarized below:

**1. Faithfulness.** PROXYSPEX learns more faithful representations of LLM outputs than marginal approaches ($\approx 15\%$ to $25\%$) on average across datasets as we vary the number of inferences. Figure 1 compares explanation faithfulness of PROXYSPEX to marginal attributions and SPEX.

**2. Feature identification.** By accounting for interactions, PROXYSPEX identifies influential features that impact model outputs more significantly than marginal approaches, and can approximate Shapley values better than KernalSHAP in the low-inference regime.

**3. Case study 1: Data attribution.** Data Attribution is the problem of identifying training points responsible for a given test prediction. On CIFAR-10 [6] PROXYSPEX identifies the interactions between training samples that most significantly impact classification performance.

**4. Case study 2: Model component attribution.** We use PROXYSPEX to study interactions between attention heads, both within and across layers, on MMLU [7] for `Llama-3.1-8B-Instruct` [8]. We observe that intra-layer interactions become more significant for deeper layers. PROXYSPEX identifies interactions that allow it to prune more heads than the LASSO.

## 2   Related work and applications

**Feature and interaction attribution.** SHAP [1] and LIME [2] are widely used for model-agnostic feature attribution. SHAP uses the game-theoretic concept of Shapley values [9] for feature attribution, while LIME fits a sparse linear model [10]. Cohen-Wang et al. [11] also consider fitting a sparse linear model for feature attribution. Chen et al. [12] uses an information-theoretic approach for feature attributions. Other methods [13, 14] study model structure to derive feature attributions.

Sundararajan et al. [15] and Bordt and von Luxburg [16] define extensions to Shapley values that consider interactions. Fumagalli et al. [17] provides a framework for computing several interaction attribution scores, but their approach does not scale past $n \approx 20$ features, which prevents them from being applied to modern ML problems that often consist of hundreds of features. Note that some feature attribution approaches such as LIME and Faith SHAP [3] are formulated explicitly as a function approximation, while others are defined axiomatically such as SHAP, though one can typically construct equivalent function approximation objectives with a suitable distance metric.

**Fourier transforms and deep learning explainability.** Several works theoretically study the spectral properties of transformers. Ren et al. [18] show transformers have sparse spectra and Hahn and Rofin [19], Abbe et al. [20] establish that they are low degree. Abbe et al. [21, 22] study the bias of networks learning interactions via a "staircase" property, i.e., using lower-order terms to learn high-order interactions. Sparsity and low degree structure is also empirically studied in [23, 24]. Kang et al. [25] shows that under sparsity in the Möbius basis [26], a representation closely related to Shapley values and the Fourier transform, interaction attributions can be computed efficiently. Mohammadi et al. [27] also learn a sparse Möbius representation for computing Shapley values. Kang et al. [4] use these insights to propose SPEX, the first robust interaction attribution algorithm to scale to the order of $n \approx 1000$ features. Gorji et al. [5] apply sparse Fourier transforms [28–31] for computing Shapley values. They also provide an algorithm to extract the Fourier transform of tree-based models using a single forward pass.

**SPEX.** We refer to the algorithm proposed in this manuscript as PROXYSPEX, in reference to SPEX, since both works exploit a sparse interaction prior to reduce computational and sample budget. SPEX uses an algebraic structured sampling scheme, coupled with error correction decoding procedures to efficiently compute the interactions in the form of a Fourier transform. In contrast, PROXYSPEX uses random samples to learn a proxy model that implicitly exploits the sparse interaction priors and our newly proposed hierarchical prior.

**Mechanistic Interpretability (MI).** MI seeks to uncover the underlying mechanisms of neural networks and transformers [32] in order to move past treating these models as *black boxes*. PROXYSPEX answers the question *"what combinations of inputs matter?"* which is a vital precursor and complement to MI investigations that subsequently address *"how does the model compute based on those specific inputs?"* Some closely related MI work attempts to recover circuits to explain underlying model behavior [33, 34]. Hsu et al. [35] use MI for interaction attribution. See Sharkey et al. [36] for a review of open problems and recent progress in MI.

## 3  PROXYSPEX

In this section, we first empirically justify our premise that significant interactions affecting LLM output are hierarchical—influential high-order interactions imply important lower-order ones. Next, we introduce PROXYSPEX, which aims to identify feature interactions for a given input $\mathbf{x}$ while minimizing the number of expensive calls to an LLM.

### 3.1  Preliminaries

**Value function.** Let $\mathbf{x}$ be the input to the LLM consisting of $n$ features[2]. For $S \subseteq [n]$, where $[n] = 1, \ldots, n$, denote $\mathbf{x}_S$ as the *masked* input where we retain features indexed in $S$ and replace all others with the [MASK] token. For example, in the sentence $\mathbf{x} =$"The sequel truly elevated the original", if $S = \{1, 2, 5, 6\}$, $\mathbf{x}_S =$ "The sequel [MASK] [MASK] the original". Masks can be more generally applied to any type of input such as image patches in a vision-language model. For a masked input $\mathbf{x}_S$ and LLM $f$, let $f(\mathbf{x}_S) \in \mathbb{R}$ denote the output of the LLM under masking pattern $S$. The value function $f$ is problem dependent. For classification tasks, a common choice is the logit of the predicted class for unmasked input, $f(\mathbf{x})$. In generative tasks, $f(\mathbf{x}_S)$ can represent the perplexity of generating the *original output* for the unmasked input. Since we focus on providing input-specific explanations, we suppress notation on $\mathbf{x}$ and denote $f(\mathbf{x}_S)$ as $f(S)$.

**Fourier transform of value function.** Let $2^{[n]}$ be the powerset of the index set. The value function $f$ can be equivalently thought of as a set function from $f : 2^{[n]} \mapsto \mathbb{R}$. Every such function admits a

---

[2]Features refer to inputs at a given granularity, e.g., tokens in an LLM or image patches in a vision model.

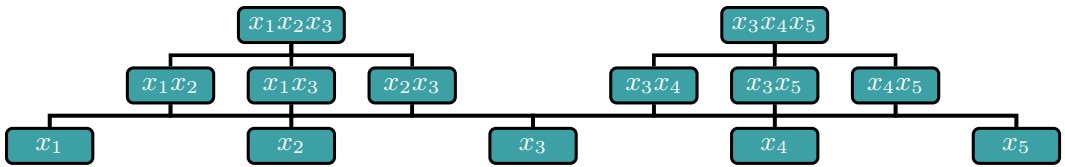

Figure 2: We observe that LLM feature interactions are often hierarchicalhigher-order interactions are accompanied by their lower-order subsets.

Fourier transform $F : 2^{[n]} \mapsto \mathbb{R}$ of $f$, related as follows:

$$\text{Transform: } F(T) = \frac{1}{2^n} \sum_{S \subseteq [n]} (-1)^{|S \cap T|} f(S), \qquad \text{Inverse: } f(S) = \sum_{T \subseteq [n]} (-1)^{|T \cap S|} F(T). \quad (1)$$

The parameters $F(T)$ are known as Fourier coefficients and capture the importance of an interaction of features in a subset $T$. Equation (1) represents an *orthonormal* transform onto a parity (XOR) basis [37]. For the rest of the paper, we use the terms Fourier coefficient and interaction interchangeably. Further, we refer to the set of Fourier coefficients $\{(T, F(T)) : T \subseteq [n]\}$ as the *spectrum*.

**Interpretable approximation of value function.** We aim to learn an interpretable approximate function $\hat{f}$ that satisfies the following:

1. **Faithful representation.** To characterize how well the surrogate function $\hat{f}$ approximates the true function, we define *faithfulness* [38]:

$$R^2 = 1 - \frac{\|\hat{f} - f\|^2}{\|f - \bar{f}\|^2}, \quad \text{where } \|f\|^2 = \sum_{S \subseteq [n]} f(S)^2, \bar{f} = \frac{1}{2^n} \sum_{S \subseteq [n]} f(S). \quad (2)$$

   Faithfulness measures how well $\hat{f}$ predicts model output. High faithfulness implies accurate approximation of $F(T)$ (this follows from orthonormality of (1)).

2. **Sparse representation.** $\hat{f}$ should be *succinct*. Previous works [4, 25, 39–41] have shown that a sparse and low-degree $\hat{f}$ can achieve high $R^2$. That is, $F(T) \approx 0$ for most $T$ (*sparsity*), and $|F(T)|$ is only large when $|T| \ll n$ (*low degree*).

3. **Efficient computation.** Without any additional assumptions on the spectrum, learning $f$ is exponentially hard since there are $2^n$ possible subsets $T$. PROXYSPEX relies on the sparse, low degree Fourier transform along with the hierarchy property to reduce LLM inferences.

A faithful and sparse $\hat{f}$ allows straightforward computation of *all* popular feature or interaction attribution scores defined in the literature, e.g., Shapley, Banzhaf, Influence Scores, Faith-Shapley. Closed-form formulas for converting $F$ to various attribution indices are provided in Appendix A.1.

## 3.2 Empirical evidence of spectral hierarchies

To quantify the degree of hierarchical structure in LLMs, we introduce the following definition called Direct Subset Rate (DSR),[3] defined for any value function $f$ and integer $k$.

$$DSR(f, k) = \frac{1}{k} \sum_{S \in \mathcal{F}_k} \frac{1}{|S|} \sum_{i \in S} \mathbb{1}\{S \setminus \{i\} \in \mathcal{F}_k\}, \quad \text{where } \mathcal{F}_k \text{ denotes the } k \text{ largest Fourier coefficients of } f. \quad (3)$$

For the top $k$ coefficients (i.e., interactions), DSR measures the average fraction of Fourier coefficients that exclude only *one* of the features $F(S \setminus \{i\})$. For example, an $f$ with $\mathcal{F}_4 = \{\emptyset, \{1\}, \{2\}, \{1, 3\}\}$ would have DSR of $\frac{1}{4}\left(1 + 1 + 1 + \frac{1}{2}\right) = \frac{7}{8}$. High DSR implies that significant high-order interactions have corresponding significant lower-order Fourier coefficients, as visualized in Figure 2. Next, we show that two LLM based value functions have high DSR.

We take 20 samples from a sentiment analysis task and an image captioning task [42]; see Section 4 for a detailed description and our choice of value function. We generate masks $S$ and apply SPEX until our learned value function has faithfulness ($R^2$) more than 0.9. Figure 3 visualizes the DSR

---

[3]For $S = \emptyset$, we set $\frac{0}{0} = 1$.

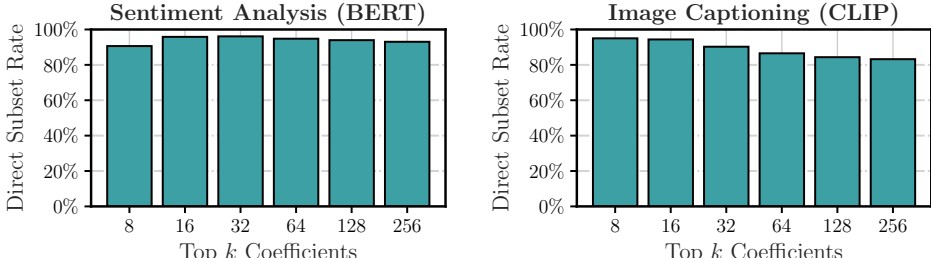

Figure 3: The top-$k$ interactions in both a sentiment analysis and image captioning task have high DSR indicating strong hierarchical structure.

for various values of $k$, i.e., number of top interactions. DSR is consistently larger than $80\%$, indicating strong hierarchical structure. In Appendix B.2, we consider two additional metrics measuring hierarchical structure, and demonstrate that the top-$k$ interactions are faithful.

**Using GBTs to capture hierarchical Interactions.** Tan et al. [43] proved that decision trees learn "staircase" functions, e.g., $f = x_1 + x_1x_2 + x_1x_2x_3$, effectively due to their greedy construction procedure. We empirically confirm this by comparing the performance of various proxy models on a synthetic hierarchical function (i.e., sum of staircase functions resembling Figure 2) as well as the Sentiment dataset in Appendix Figure 13. Appendix B.4 details the simulation set-up. GBTs vastly outperform other proxy models, indicating their natural ability to identify hierarchical interactions with limited training data. Interestingly, GBTs outperform random forests as well. This is because random forests are ineffective at learning hierarchical functions [44], i.e., sums of staircases, while GBT-like algorithms disentangle sums effectively [45].

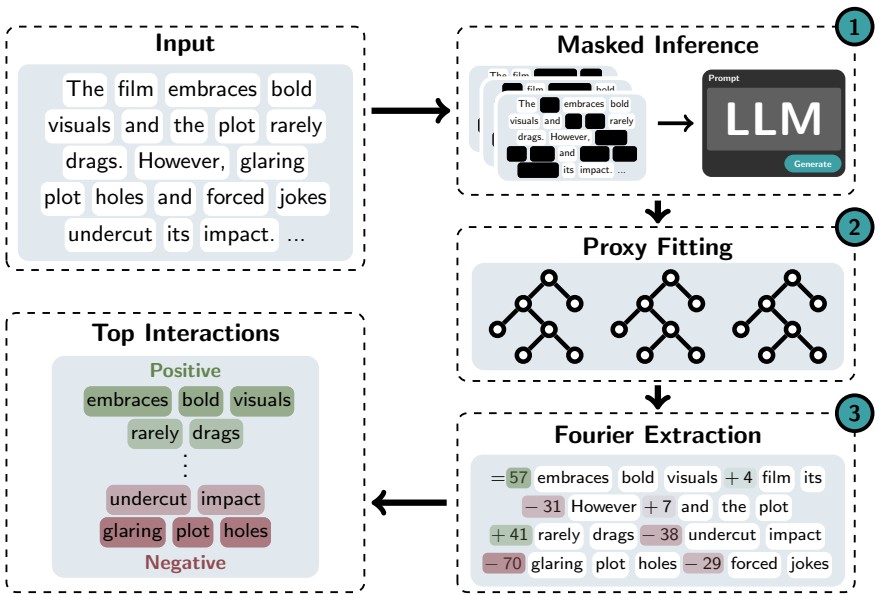

Figure 4: (1) PROXYSPEX masks subsets of words and queries the LLM using this masked input. (2) It then fits GBTs as a proxy model to learn the LLM's hierarchical interactions. (3) An interpretable sparse representation is extracted from the fitted GBT which captures the influential interactions.

## 3.3 PROXYSPEX via Gradient Boosted Trees to fit hierarchies

The PROXYSPEX algorithm (see Figure 4):

**Step 1 - Sampling and querying.** Given LLM $f$ and input instance $\mathbf{x}$ to explain, generate a dataset $\mathcal{D} = (S_i, f(S_i))_{i=1}^{\ell}$ for training the proxy. The inputs $S_i$ represent the masks of $\mathbf{x}$. Each mask $S_i$ is sampled uniformly from the set $[n]$. The labels $f(S_i)$ are obtained by querying the LLM.

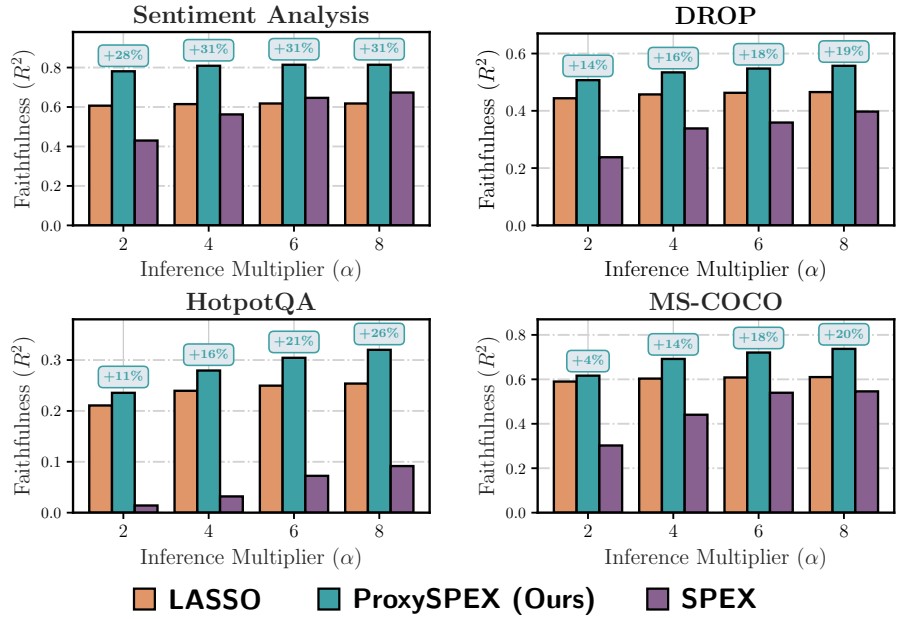

Figure 6: Comparison of faithfulness of different attribution methods with $\alpha \cdot n \log_2(n)$ training masks for different inference multipliers $\alpha \in \{2, 4, 6, 8\}$. While SPEX is only competitive with LASSO for large $\alpha$, the gap between PROXYSPEX and LASSO increases with $\alpha$.

**Step 2 - Proxy Training.** Fit GBTs to $\mathcal{D}$ with 5-fold cross-validation (CV).

**Step 3 - Fourier extraction.** We use Gorji et al. [5] to extract the Fourier representation of the fitted GBTs in a single forward pass; see Appendix A.2. With $T$ trees of depth $d$ there are at most $O(T4^d)$ non-zero Fourier coefficients [5]. To improve interpretability, we sparsify the extracted representation by keeping only the top $k$ Fourier coefficients. Fig. 5 shows that only $\approx 200$ Fourier coefficients are needed to achieve equivalent faithfulness for a sentiment classification and image captioning (MS-COCO) dataset. Additional results regarding the sparsity of Fourier spectra learned by GBTs are in Appendix B.3.

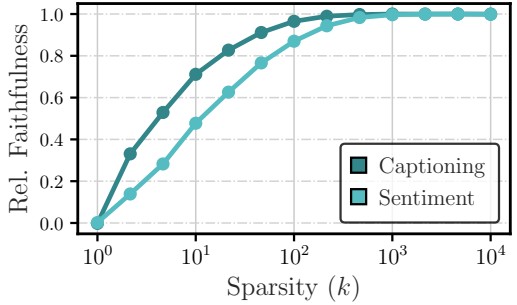

Figure 5: Relative faithfulness as a function of Fourier sparsity. Only $\approx 200$ coefficients are required to achieve equivalent faithfulness. Sparsity for sentiment is higher since inputs have larger $n$.

**Step 4 (Optional): Coefficient refinement via regression.** As a final step, we optionally regress the extracted, top $k$ Fourier coefficients on the collected data $\mathcal{D}$ to improve the estimation. Empirically we observe this step is can sometimes marginally improve performance, but seldom negatively impacts performance. This step is included if it leads to lower CV error.

## 4    Results

**Datasets and models**

1. *Sentiment* is a classification task composed of the *Large Movie Review Dataset* [46] which consists of positive and negative IMDb movie reviews. We use words as input features and restrict to samples with $n \in [256, 512]$. We use the encoder-only fine-tuned `DistilBERT` model [47, 48], and the logit of the positive class as the value function.

2. *HotpotQA* [49] is a generative question-answering task over Wikipedia articles. Sentences are input features, and we restrict to samples with $n \in [64, 128]$. We use `Llama-3.1-8B-Instruct`, and perplexity of the unmasked output as the value function.

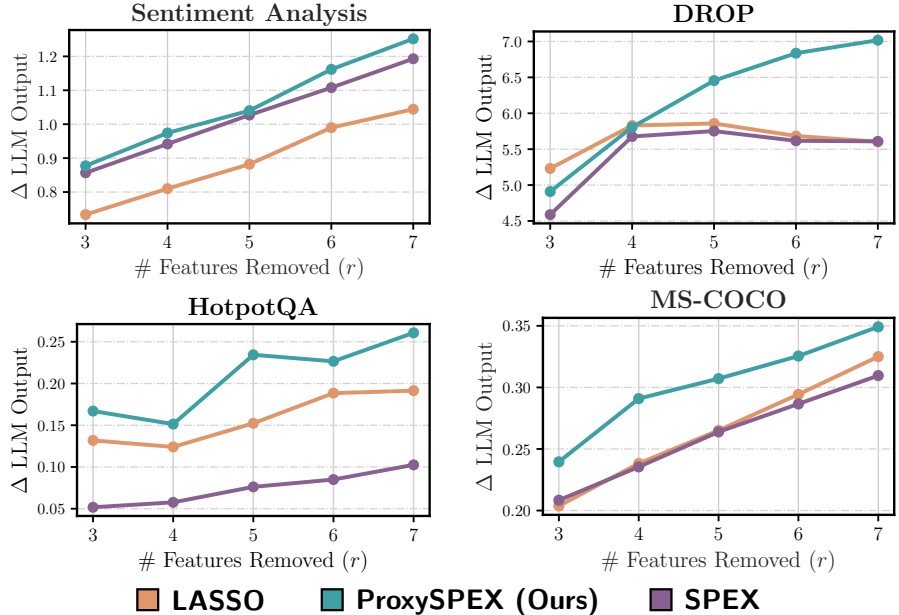

Figure 7: By accounting for interactions, PROXYSPEX identifies more influential features across datasets than the LASSO. Apart from the sentiment analysis task (top left), SPEX does not collect enough training masks to out-perform LASSO.

3. *Discrete Reasoning Over Paragraphs* (DROP) [50] is a paragraph level question-answering task. We use words as input features and restrict to samples with $n \in [256, 512]$. We use `Llama-3-8B-Instruct` and the perplexity of the unmasked output as the value function.

4. *MS-COCO* [42] contains images and corresponding text captions. Image patches and words are the input features with $n \in [60, 85]$. We use `CLIP-ViT-B/32`, a joint vision-language encoder, with the value function defined as the contrastive loss over all datapoints.

**Baselines and hyperparameters.** For marginal feature attributions, we use the LASSO. We use the same datasets at [4] and add *MS-COCO* for an additional modality. It was shown in [4] that popular marginal metrics such as SHAP are significantly less faithful than the LASSO, e.g., have $R^2 < 0$. We use the LASSO implementation from `scikit-learn`, and choose the $l_1$ regularization parameter via 5-fold CV. For interaction indices, we compare PROXYSPEX to SPEX. Due to the scale of $n$ in our experiments, we cannot compare methods for computing interaction indices such as Faith-Shapley, Faith-Banzhaf, and Shapley-Taylor using SHAP-IQ [17], and SVARM-IQ [51], because they enumerate all possible interactions, making them computationally infeasible. For PROXYSPEX, a list of GBT hyper-parameters we tune over are in Appendix B.

## 4.1 Faithfulness

We compare attribution method faithfulness by varying the number of training masks. For each sample with $n$ features, we generate $\alpha \cdot n \log_2(n)$ masks, varying $\alpha \in \{2, 4, 6, 8\}$, to normalize difficulty across inputs of varying lengths (some by over 100 tokens). This $n \log(n)$ type scaling is heuristically guided by compressed sensing bounds [52]. These suggest the number of samples required grows with sparsity (assumed $\propto n$) and logarithmically with problem dimensionality (if dimensionality for degree-$d$ interactions is $\approx n^d$, this yields a $\log(n^d) = d \log(n)$ factor). Together, these factors support an $n \log(n)$ scaling. While not directly applicable, these bounds offer a useful heuristic for how sampling complexity scales with $n$.

Figure 6 shows average faithfulness over 1,000 test masks per sample. PROXYSPEX outperforms LASSO with limited inferences and continues to improve where LASSO plateaus, indicating that it is learning influential interactions. While SPEX is often faster for the same number of masks, SPEX needs additional inference time to match $R^2$, making PROXYSPEX faster overall. For the smaller `DistilBERT` model under the sentiment analysis task, the wall clock speedup is $\sim 3\times$, while with the bigger `CLIP-ViT-B/32` model with MS-COCO we see $\sim 5\times$ speedup (See Appendix B.6).

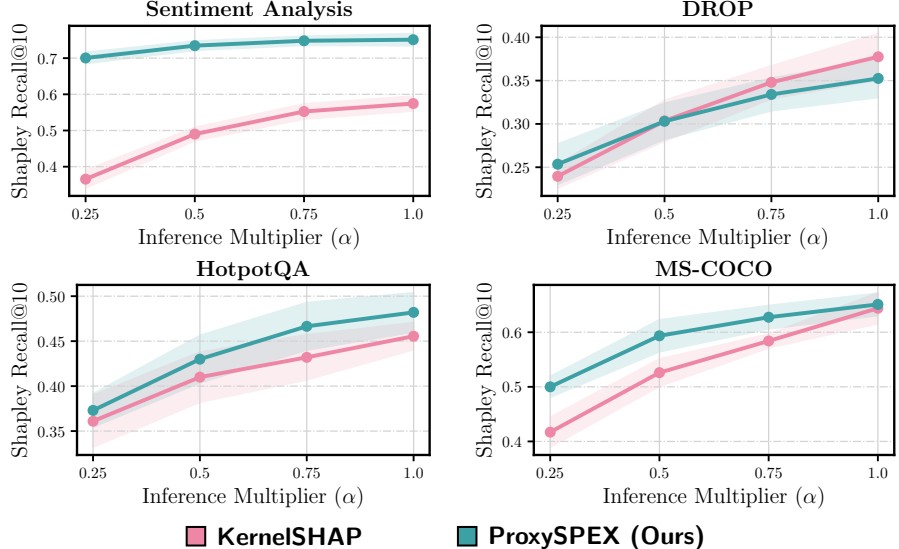

Figure 8: Recall of the top ten Shapley values after $\alpha \cdot n \log_2(n)$ inferences for multipliers $\alpha \in \{0.25, 0.5, 0.75, 1.0\}$. For small $\alpha$, PROXYSPEX is superior at recovering the most significant features, while KernalSHAP outperforms as $\alpha$ increases. Error bands indicate the standard deviation across ten different runs of the algorithms.

## 4.2 Feature Identification

We measure the ability of methods to identify the top $r$ features influencing LLM outputs:

$$\Delta \text{ LLM Output } (r) = \frac{|f([n]) - f(S^*)|}{|f([n])|}, \quad S^* = \underset{|S|=n-r}{\operatorname{argmax}} |\hat{f}([n]) - \hat{f}(S)|. \tag{4}$$

Solving Eq. 4 for an arbitrary $\hat{f}$ presents a challenging combinatorial optimization problem. However, PROXYSPEX and SPEX represent $\hat{f}$ as a sparse Fourier transform. This representation facilitates solving the optimization as a tractable linear integer program. The sparsity of the extracted Fourier representation ensures that the time required to solve this program is negligible compared to sampling the LLM and fitting the GBTs. Full details of the construction of this program are given in Appendix A.3. Under LASSO, Eq. 4 is easily solved through selecting features by the size of their coefficients. We measure the removal ability of different attribution methods when we collect $8n \log_2(n)$ training masks and plot the result in Figure 7. By accounting for interactions, PROXYSPEX identifies significantly more influential features than the LASSO. Apart from the sentiment analysis task, SPEX does not collect enough training masks to outperform the LASSO.

## 4.3 Shapley Value Approximation

PROXYSPEX can be directly used to approximate Shapley values. Across all tasks, we first run KernelSHAP with 10,000 test masks and treat these approximated Shapley values as ground truth. We measure the recall of the top ten highest-magnitude Shapley values for KernelSHAP and PROXYSPEX under $\alpha \cdot n \log_2(n)$ inferences with multipliers $\alpha \in \{0.25, 0.5, 0.75, 1.0\}$. For this inference budget, competing algorithms such as LeverageSHAP [53] and SVARM [54] struggle to provide accurate approximations. We find PROXYSPEX initially provides a better coarse approximation than KernelSHAP (Figure 14). However, since PROXYSPEX is optimized for faithfulness and does not rely on the Shapley kernel, it is eventually surpassed by KernelSHAP with enough inferences. Additional results under mean squared error are included in Appendix B.5.

## 5 Case studies

We now present two case studies of PROXYSPEX for two different interpretability problems: *data attribution* [55] and *model component attribution* [56], a key problem in mechanistic interpretability. We first show how both of these tasks can be reformulated as feature attribution tasks; recent work has highlighted the connections between feature, data, and model component attribution [57].

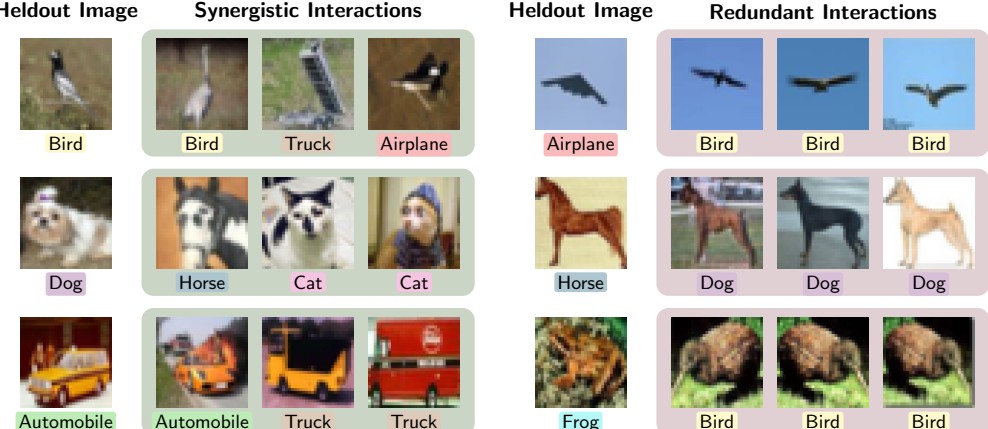

Figure 9: Synergistic interactions: data that together are more valuable together than the sum of their parts and aid in classification. Redundant interactions: Data that may contain similar information, their combined influence is less than the sum of the parts.

## 5.1 Data Attribution via Non-Linear Datamodels

Data attribution for classification is the problem of understanding how fitting a model $g_\theta$ on a subset $S$ of training samples affects the prediction of a test point $\mathbf{z}$ of class $c$. This problem can be converted into our framework by defining an appropriate value function $f$,

$$f(S) \triangleq (\text{logit for } c \text{ on } \mathbf{z}) - (\text{highest incorrect logit on } \mathbf{z}), \quad \text{when } g_\theta \text{ is trained on } S. \quad (5)$$

The value function $f$ quantifies the impact of a subset $S$ on the classification of $\mathbf{z}$. Sampling $f$ is very expensive since it involves training a new model $g_\theta$ for every subset $S$. As a result, most data attribution approaches do not consider the impact of interactions. Notably, Ilyas et al. [55] use LASSO to learn $f$ when training a ResNet model on the CIFAR-10 dataset [6]. As a case study, we apply PROXYSPEX to understand the impact of interactions between CIFAR-10 training samples.

**Defining data interactions.** Interactions between samples can be either *redundant interactions* or *synergistic interactions*. Redundant interactions are when the influence of a subset $S$ is not additive. Redundancy typically occurs between highly correlated samples, e.g., semantic duplicates [58]. Synergistic interactions occur when a subset $S$ influences a prediction by shaping a decision boundary that no individual sample in $S$ could do so by itself. That is, the model needs the combined effect of training samples in $S$ to correctly classify $\mathbf{z}$.

**Results.** We visualize interactions learned by PROXYSPEX in Figure 9 for randomly selected CIFAR-10 test points. Experimental details are in Appendix C.1. PROXYSPEX identifies highly similar training samples (redundancies) as well as synergistic interactions between samples of different classes. See Appendix C.1 for examples of other randomly selected test samples.

## 5.2 Model Component Attribution

We study the role of attention heads for a question-answering task using `Llama-3.1-8B-Instruct` and MMLU (high-school-us-history), which is a multiple-choice dataset. We treat each attention head as a feature and aim to identify interactions among heads using PROXYSPEX. Let $L$ represent the number of layers in an LLM and let $\mathcal{L} \subseteq [L]$ represent a subset of the layers. Let $\mathcal{H}_\mathcal{L}$ denote the set of attention heads within these layers. For a subset of heads $S \subseteq \mathcal{H}_\mathcal{L}$, we set the output of heads in $\mathcal{H}_\mathcal{L} \setminus S$ to 0 and denote the ablated LLM as $\text{LLM}_S(\cdot)$. Define $f$ as:

$$f_\mathcal{L}(S) \triangleq \text{Accuracy of } \text{LLM}_S \text{ on training set of MMLU.} \quad (6)$$

**Pruning results.** We use the LASSO and PROXYSPEX to identify the most important heads for various sparsity levels ( i.e., the number of retained heads) across different sets of layers. We also compare to a Best-of-$N$ baseline, where we take the best of $N = 5000$ different randomly chosen $S$, further details are in Appendix C.2. We use the procedure detailed in Section 4.2 to identify heads

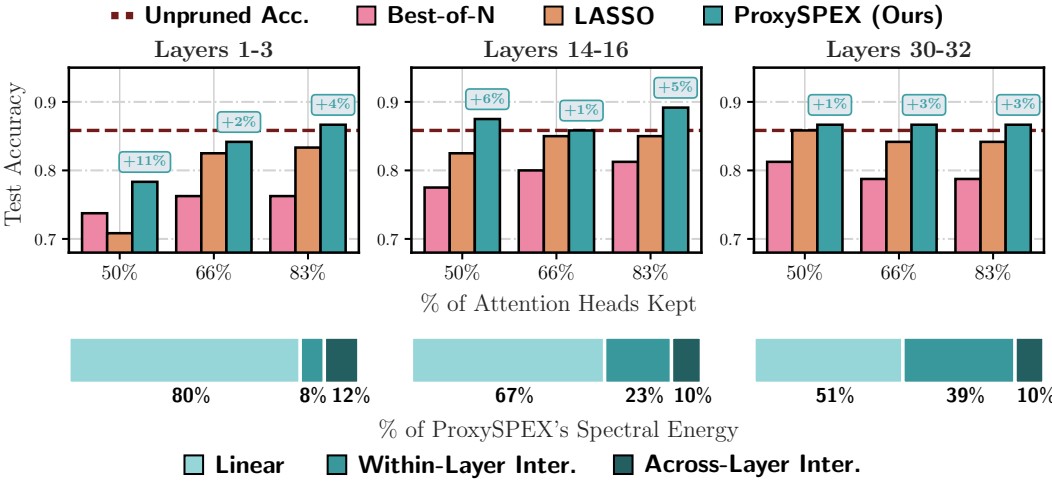

Figure 10: Attention head pruning for `Llama-3.1-8B-Instruct` for MMLU (high-school-us-history). **Top:** We report the test accuracy vs. percentage of heads retained, comparing PROX-YSPEX, LASSO, and Best-of-$N$ across layer groups (1-3, 14-16, 30-32). Unpruned accuracy shown by dashed line. **Bottom:** PROXYSPEX's learned spectral energy distribution into linear effects, within-layer, and across-layer interactions per layer group.

to remove for both PROXYSPEX and LASSO. Test accuracies for each method are presented in Figure 10 at three different sparsity levels, and with three different layer ranges: initial (1-3), middle (14-16) and final (30-32). We observe that PROXYSPEX consistently outperforms both baselines, with a higher test accuracy on the pruned models identified using PROXYSPEX.

**Characterizing interactions between attention heads.** Analyzing the Fourier spectrum learned by PROXYSPEX offers insights into the nature of the internal mechanisms of the LLM. As shown in Figure 10 (bottom), the spectral energy attributed to interactions, particularly *within-layer* interactions, markedly increases in deeper layers of `Llama-3.1-8B-Instruct`. There are many works that look at the differing functional roles of attention heads across layers [59]. PROXYSPEX provides an exciting new quantitative approach to further investigate these phenomena.

# 6 Discussion

**Conclusion.** We introduce PROXYSPEX, an inference-efficient interaction attribution algorithm that efficiently scales with $n$ by leveraging an observed hierarchical structure among significant interactions in the Fourier spectrum of the model. Experiments across 4 high-dimensional datasets show that PROXYSPEX exploits hierarchical interactions via a GBT proxy model to reduce inferences by $\sim 10\times$ over SPEX [4] while achieving equally faithful explanations. Through applications to data and model component attribution, we demonstrate the importance of efficient interaction discovery.

**Limitations.** GBTs effectively capture hierarchical interactions but may not perform as well when interactions have a different structure. For example, simulations in Appendix B.4 empirically confirm that GBTs suffer in the case of sparse but non-hierarchical functions. More generally, in cases where the proxy GBT model is not faithful, the interactions identified by PROXYSPEX might not be representative of the model's reasoning. Another limitation is the degree of human interpretability that can be understood from computed interactions. While interactions can offer richer insights, they are more difficult to parse than marginal alternatives. Further improvements in visualization and post-processing of interactions are needed to fully harness the advances of PROXYSPEX.

**Future work.** Inference-efficiency could be further improved by exploring alternative proxy models, additional Fourier spectral structures, or adaptive masking pattern designs. Integrating PROX-YSPEX with internal model details, such as via hybrid approaches with MI or by studying its connection to sparsity in transformer attention [60], offers another promising avenue. Finally, further deepening and improving applications of PROXYSPEX in data attribution and mechanistic interpretability as well as potentially exploring more complex value functions or larger-scale component interactions remains interesting future work.

## Acknowledgments and Disclosure of Funding

This material is based upon work supported by the National Science Foundation Graduate Research Fellowship Program under Grant No. DGE-2146752. Any opinions, findings, and conclusions or recommendations expressed in this material are those of the author(s) and do not necessarily reflect the views of the National Science Foundation.

This work used NCSA DeltaAI at UIUC through allocation CIS250245 from the Advanced Cyberinfrastructure Coordination Ecosystem: Services & Support (ACCESS) program, which is supported by U.S. National Science Foundation grants #2138259, #2138286, #2138307, #2137603, and #2138296.

B.Y. gratefully acknowledge partial support from NSF grant DMS-2413265, NSF grant DMS 2209975, NSF grant 2023505 on Collaborative Research: Foundations of Data Science Institute (FODSI), the NSF and the Simons Foundation for the Collaboration on the Theoretical Foundations of Deep Learning through awards DMS-2031883 and 814639, NSF grant MC2378 to the Institute for Artificial CyberThreat Intelligence and OperatioN (ACTION), and NIH grant R01GM152718.

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

# Appendices

# A  Method Details

## A.1  Fourier Conversions

| INTERACTION INDEX | FOURIER CONVERSION |
|---|---|
| Banzhaf $\psi_i$ | $-2F(\{i\})$ |
| Shapley $\phi_i$ | $(-2)\sum_{\substack{S \supseteq \{i\} \\ \|S\| \text{ is odd}}} \frac{F(S)}{\|S\|}$ |
| Influence $\xi_i$ | $\sum_{S \ni i} F(S)^2$ |
| Möbius $I^{\mathrm{M}}(T)$ | $(-2)^{\|T\|} \sum_{S \supseteq T} F(S)$ |
| Or $I^{\mathrm{O}}(T)$ | $\begin{cases} \sum_{S \subseteq [n]} F(S) & \text{if } T = \emptyset \\ -(-2)^{\|T\|} \sum_{S \supseteq T} (-1)^{\|S\|} F(S) & \text{if } T \neq \emptyset \end{cases}$ |
| Banzhaf Interaction $I^{\mathrm{B}}(T)$ | $-2F(T)$ |
| Shapley Interaction $I^{\mathrm{S}}(T)$ | $(-2)^{\|T\|} \sum_{S \supseteq T \; s.t. \; (-1)^{\|S\|}=(-1)^{\|T\|}} \frac{F(S)}{\|S\|-\|T\|+1}$ |
| Shapley Taylor $I^{\mathrm{ST}}_\ell(T)$ | $\begin{cases} I^{\mathrm{M}}(T), & \|T\| < \ell, \\ \sum_{S \supseteq T} \binom{\|S\|}{\ell}^{-1} I^{M}(S), & \|T\| = \ell. \end{cases}$ |
| Faith-Banzhaf $I^{\mathrm{FB}}_\ell(T)$ | $(-2)^{\|T\|} \sum_{\substack{S \supseteq T \\ \|S\| \leq \ell}} F(S)$ |
| Faith-Shapley $I^{\mathrm{FS}}_\ell(T)$ | $I^{M}(T) + (-1)^{\ell-\|T\|} \frac{\|T\|}{\ell+\|T\|} \binom{\ell}{\|T\|} \sum_{\substack{S \supseteq T \\ \|S\|>\ell}} F(S)\, \gamma(S,T,\ell)$ 
 $\text{where } \gamma(S,T,\ell) = \sum_{\substack{T \subset R \subseteq S \\ \|R\|>\ell}} \frac{\binom{\|R\|-1}{\ell}}{\binom{\|R\|+\ell-1}{\ell+\|T\|}} (-2)^{\|R\|}$ |

The relationship between Fourier coefficients and influence scores are provided in [37]. We derive the conversion between Fourier and the OR interaction index [61] in this work. All remaining conversions are derived in Appendix C of [4].

## A.2  Fourier Extraction

The exact Fourier transform of a decision tree can be computed recursively [5, 62, 63]. Due to the linearity of the Fourier transform, the Fourier transform of each boosted tree can be computed separately and added together. Algorithm 1, provided by [5], proceeds by traversing the nodes of each tree and summing the resultant Fourier transforms.

---

**Algorithm 1** Fourier Extraction from Gradient Boosted Trees [5]

---

**Require:** Gradient boosted model $\mathcal{M}$
**Ensure:** Fourier mapping $\mathcal{F}$
 1: Initialize $\mathcal{F} \leftarrow \emptyset$
 2: **for** Tree $T$ in $\mathcal{M}$ **do**
 3:     $\mathcal{F} \leftarrow \mathcal{F}.\texttt{merge}(\text{EXTRACTTREE}(T.\texttt{root}))$        ▷ Add mappings of the individual trees
 4: **end for**
 5: **return** $\mathcal{F}$

 6: **procedure** EXTRACTTREE(node $n$)
 7:     **if** $n$ is leaf **then**
 8:         **return** $\{\emptyset \mapsto n.\texttt{value}\}$
 9:     **else**
10:         $\mathcal{N}_L \leftarrow \text{EXTRACTTREE}(n.\texttt{leftChild})$
11:         $\mathcal{N}_R \leftarrow \text{EXTRACTTREE}(n.\texttt{rightChild})$
12:         $\mathcal{N} \leftarrow \emptyset$
13:         **for** $S$ in $(\mathcal{N}_L.\texttt{keys} \cup \mathcal{N}_R.\texttt{keys})$ **do**
14:             $v_L \leftarrow \mathcal{N}_L[S]$        ▷ Mapping returns 0 if not contained
15:             $v_R \leftarrow \mathcal{N}_R[S]$
16:             $\mathcal{N}[S] \leftarrow (v_L + v_R)/2$
17:             $\mathcal{N}[S \cup \{n.\texttt{featureSplit}\}] \leftarrow (v_L - v_R)/2$
18:         **end for**
19:     **end if**
20:     **return** $\mathcal{N}$
21: **end procedure**

---

## A.3 Sparse Fourier Optimization

We assume $\hat{f}(S)$ is a sparse, low-degree function with support $\mathcal{K}$:

$$\hat{f}(S) = \sum_{T \in \mathcal{K}} (-1)^{|S \cap T|} \hat{F}(T)$$

Equivalently, the function can be represented (and efficiently converted) under the Möbius transform. Converting Fourier to Möbius (via Appendix A.1), letting $\mathcal{K}^+ = \big\{ R \subseteq T \mid T \in \mathcal{K} \big\}$, and applying the inverse Möbius transform:

$$\hat{f}(S) = \sum_{R \in \mathcal{K}^+, R \subseteq T} \hat{I}^M(R)$$

The optimization problem can then be expressed as a polynomial over $\{0,1\}$. Let $\mathbf{x}$ be a binary vector of length $n$ and $S = \{i \in [n] \mid x_i = 1\}$. We will focus on the maximization problem (minimization follows analogously).

$$\max_{S \subseteq [n]} \hat{f}(S) = \max_{\mathbf{x} \in \{0,1\}^n} \sum_{R \in \mathcal{K}^+} \hat{I}^M(R) \prod_{i \in R} x_i$$

To reduce the problem to a linear integer program, each monomial $\prod_{i \in R} x_i$ can be replaced with a decision variable $y_R$ and the following constraints:

$$\max_{\mathbf{y} \in \{0,1\}^{|\mathcal{K}^+|}} \sum_{R \in \mathcal{K}^+} \hat{I}^M(R) y_R \tag{7}$$

$$\text{s.t.} \quad y_R \leq y_Q \qquad \forall\, Q \subset R,\ R \in \mathcal{K}^+ \tag{8}$$

$$\sum_{i \in R} y_{\{i\}} < |R| + y_R \quad \forall\, R \in \mathcal{K}^+ \tag{9}$$

The first constraint guarantees that whenever a monomial is activated (i.e. $x_i = 1\ \forall i \in R$), all of its subsets are also activated. The second constraint ensures that if a monomial is deactivated (i.e. $\exists\, i \in R\ s.t.\ x_i = 0$), at least one of its constituent terms ($y_{\{i\}}$) is likewise deactivated. After

the optimization is solved, the solution can be read-off from the univariate monomials $y_{\{i\}}$. These monomial terms can also be used to impose cardinality constraints on the solution, as was used in Section 4.2 and Section 5.2.

# B  Experimental Details

## B.1  Implementation Details

### B.1.1  Hyper-parameters

We performed 5-fold cross-validation over the following hyper-parameters for each of the models:

| Model | Hyper-parameter |
|---|---|
| LASSO | L1 Reg. Param. $\lambda$ (100 with $\lambda_{min}/\lambda_{max} = 0.001$) |
| SPEX | L1 Reg. Param. $\lambda$ (100 with $\lambda_{min}/\lambda_{max} = 0.001$) |
| PROXYSPEX | Max. Tree Depth [3, 5, None] |
| | Number of Trees [500, 1000, 5000] |
| | Learning Rate [0.01, 0.1] |
| | L1 Reg. Param. $\lambda$ (100 with $\lambda_{min}/\lambda_{max} = 0.001$) |
| Random Forest | Max. Tree Depth [3, 5, None] |
| | Number of Trees [100, 500, 1000, 5000] |
| Neural Network | Hidden Layer Sizes $[(\frac{n}{4}), (\frac{n}{4}, \frac{n}{4}), (\frac{n}{4}, \frac{n}{4}, \frac{n}{4})]$ |
| | Learning Rate [Constant, Adaptive] |
| | Learning Rate Init. [0.001, 0.01, 0.1] |
| | Number of Trees [100, 500, 1000, 5000] |

### B.1.2  Sentiment Analysis

20 movie reviews were used from the *Large Movie Review Dataset* [46] with $n \in [256, 512]$ words. To measure the sentiment of each movie review, we utilize a `DistilBERT` model [47] fine-tuned for sentiment analysis [48]. When masking, we replace the word with the `[UNK]` token. We construct an value function over the output logit associated with the positive class.

### B.1.3  HotpotQA

We consider 50 examples from the *HotpotQA*[49] dataset between $n \in [64, 128]$ sentences. We use a `Llama-3.2-3B-Instruct` model with 8-bit quantization. When masking, we replace with the `[UNK]` token, and measure the log-perplexity of generating the original output. Since *HotpotQA* is a multi-document dataset, we use the following prompt format.

> **Title:** {title_1}
> **Content:** {document_1}
> . . .
> **Title:** {title_m}
> **Content:** {document_m}
>
> **Query:** {question}. Keep your answers as short as possible.

### B.1.4  DROP

We consider 50 examples from the *DROP* [49] dataset with $n \in [256, 512]$ number of words. We use the same model as *HotpotQA* and mask in a similar fashion. We use the following prompt format.

> **Context:** {context}
> **Query:** {question}. Keep your answers as short as possible.

### B.1.5 MS-COCO

We utilize the Microsoft Common Objects in Context (MS-COCO) dataset [42], which comprises images paired with descriptive text captions. For our experiments, we treat image patches (there are $48$ patches per image) and individual words from the captions as the input features. We used the first $50$ examples from the test set, which had $n$ (image patches + words) between the range of $[60, 85]$.

To model the relationship between images and text, we employed the `CLIP-ViT-B/32` model, a vision-language encoder designed to learn joint representations of visual and textual data. In our PROXYSPEX framework, when masking input features (either image patches or words), we replace them with a generic placeholder token suitable for the CLIP architecture (e.g., a zeroed-out patch vector or the text `[MASK]` words). The value function $f(S)$ for a given subset of features $S$ was defined as the contrastive loss among the other image/caption pairs. By measuring the change in this contrastive loss upon masking different feature subsets, we can attribute importance to individual features and their interactions in the context of joint image-text understanding.

### B.2 Measuring Spectral Hierarchies

To quantify the hierarchical structure observed in the Fourier spectra of the LLMs under study, we introduce and analyze two key metrics: the Staircase Rate ($SCR$) and the Strong Hierarchy Rate ($SHR$). These metrics are computed based on the set of the k largest (in magnitude) Fourier coefficients, denoted as $\mathcal{F}_k$.

The *Staircase Rate* ($SCR(f, k)$) is defined as:

$$SCR(f,k) = \frac{1}{k} \sum_{S \in \mathcal{F}_k} \mathbb{1} \left\{ \exists (e_1, \ldots, e_{|S|}) \in \text{Perm}(S) \text{ s.t. } \left( \forall j \in \{0, \ldots, |S|\} : \bigcup_{l=1}^{j} \{e_l\} \in \mathcal{F}_k \right) \right\},$$

where $\mathcal{F}_k$ denotes the $k$ largest Fourier coefficients of $f$,
and $\text{Perm}(S)$ is the set of all ordered sequences of the elements in $S$.

(10)

The $SCR$ measures the proportion of top-$k$ Fourier coefficients $F(S)$ for which there exists an ordering of its constituent elements $(e_1, \ldots, e_{|S|})$ such that all initial subsets (i.e., $e_1, \{e_1, e_2\}, \ldots, S$ itself) are also among the top-$k$ coefficients. A high $SCR$ indicates that significant high-order interactions are built up from significant lower-order interactions in a step-wise or "staircase" manner.

The *Strong Hierarchy Rate* ($SHR(f, k)$) is defined as:

$$SHR(f,k) = \frac{1}{k} \sum_{S \in \mathcal{F}_k} \mathbb{1} \left\{ \forall S' \subseteq S, S' \in \mathcal{F}_k \right\}, \quad \text{where } \mathcal{F}_k \text{ denotes the } k \text{ largest Fourier coefficients of } f. \quad (11)$$

The $SHR$ is a stricter measure, quantifying the proportion of top-$k$ coefficients $F(S)$ for which all subsets of $S$ (not just initial subsets, as in $DSR$) are also present in $\mathcal{F}_k$. A high $SHR$ suggests a very robust hierarchical structure where the significance of an interaction implies the significance of all its underlying components.

Figure 11 visualizes these rates alongside faithfulness ($R^2$) for the Sentiment Analysis and MS-COCO datasets. These empirical results aim to demonstrate that LLM feature interactions exhibit significant hierarchical structure. The high $SCR$ and $SHR$ scores support the core motivation for PROXYSPEX: that important interactions are often built upon their lower-order subsets, a structure that Gradient Boosted Trees (GBTs) are well-suited to capture and exploit.

### B.3 Sparsification

The process of sparsification is crucial for enhancing the interpretability of the explanations generated by PROXYSPEX . By retaining only the top $k$ Fourier coefficients, we can achieve a more concise and understandable representation of the model's behavior without significantly compromising the faithfulness of the explanation. As demonstrated in Figure 5, a relatively small number of Fourier coefficients (approximately 200) are often sufficient to achieve faithfulness comparable to using a much larger set of coefficients for tasks like sentiment classification and image captioning (MS-COCO).

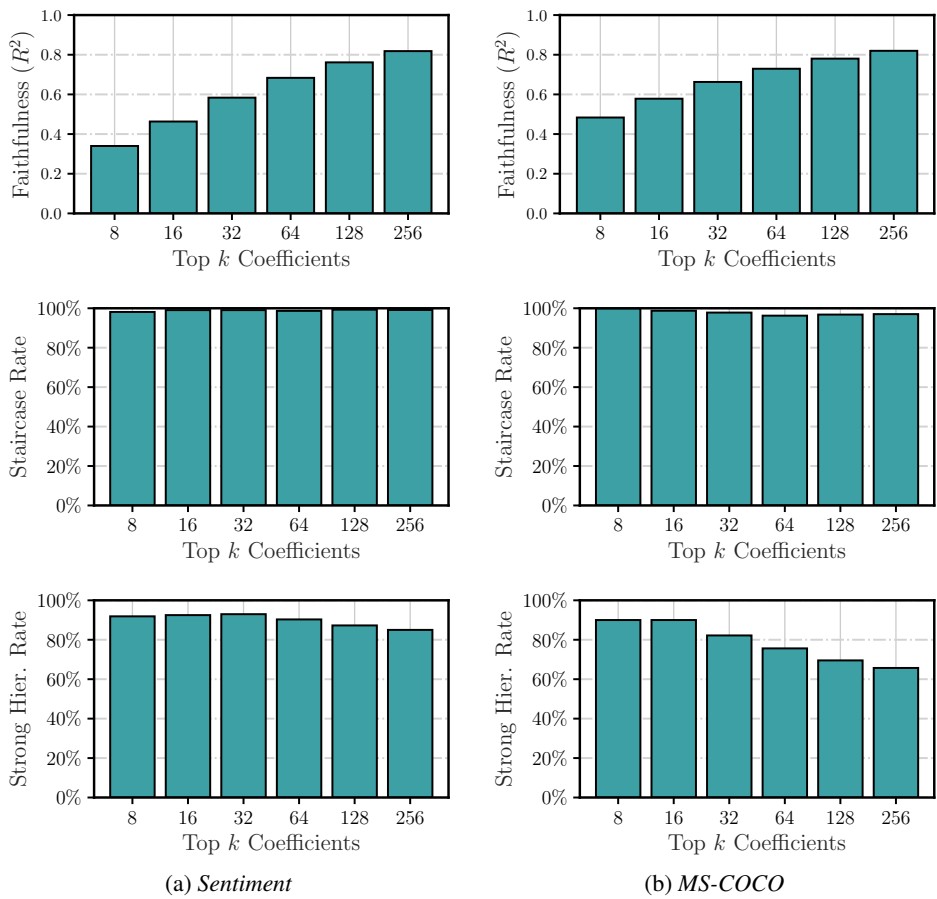

(a) *Sentiment*                    (b) *MS-COCO*

Figure 11: (top row) We run SPEX until $R^2 > 0.9$. We report the faithfulness of when we truncate the spectrum to keep just the top $k$ coefficients for a range of $k$. We include results from Sentiment $n \in [256, 512]$, and MS-COCO $n \in [60, 85]$. In both cases faithfulness steadily increases as we increase $k$. (middle row) We report the $SCR$ (10) for the same top $k$ Fourier truncated functions above. In all cases, the $SCR$ is nearly 100%. (bottom row) We also report the $SHR$ (11), which is the strongest of the metric we consider. Here we find that even though $SHR$ decreases somewhat as $k$ grows, it is still strongly in favor of the hierarchy hypothesis.

Further results in Figure 12 illustrate the relationship between relative faithfulness and Fourier sparsity for both Sentiment and MS-COCO datasets across different inference multipliers ($\alpha$). These plots show that faithfulness generally increases with $k$, plateauing after a certain number of coefficients, reinforcing the idea that a sparse representation can effectively capture the essential dynamics of the LLM's decision-making process.

## B.4 Proxy Model Selection

The choice of GBTs as the proxy model within PROXYSPEX is motivated by their inherent ability to identify and learn hierarchical interactions from limited training data. This is a critical characteristic, as LLM feature interactions often exhibit a hierarchical structure where higher-order interactions are built upon their lower-order subsets. As indicated in the main text, GBTs have been shown to vastly outperform other proxy models, including random forests, particularly because random forests are less effective at learning hierarchical functions. GBT-like algorithms, on the other hand, are adept at disentangling sums of these hierarchical components.

Figure 13 provides a comparative view of proxy model performance. Figure 13a and Figure 13b illustrate the faithfulness ($R^2$) of different proxy models (LASSO, Random Forest, Neural Network,

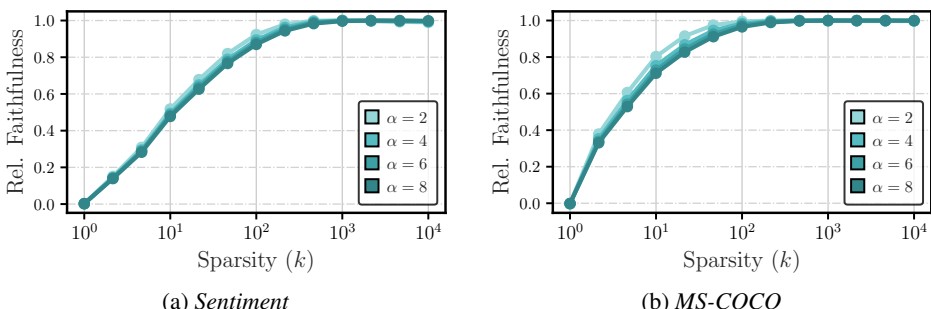

| (a) *Sentiment* | (b) *MS-COCO* |

Figure 12: We plot faithfulness ($R^2$) as a function of Fourier sparsity. Only $\approx 200$ coefficients are required to achieve equivalent faithfulness.

and GBTs) on both a synthetic dataset with a complete hierarchy (defined below) and the Sentiment Analysis dataset, respectively, across various inference parameters ($\alpha$). These results empirically support the superiority of GBTs in capturing these complex interaction structures. However, it's also important to acknowledge limitations; for instance, GBTs may not perform as well when interactions possess a different, non-hierarchical sparse structure, as empirically confirmed by simulations like the Synthetic-Peak example (which lacks hierarchical structure) shown in Figure 13c.

| Synthetic Peak | Synthetic Complete Hierarchy |
|---|---|
| $f^{\text{SP}}(S) = \sum_{T \subseteq \mathcal{P}} (-1)^{\|S \cap T\|} F(T)$ | $f^{\text{SCH}}(S) = \sum_{R \subseteq \mathcal{H}} (-1)^{\|S \cap R\|} F(R)$ |
| where $\mathcal{P}$ is a set of 10 uniformly | where $\mathcal{H} = \{R \subseteq T \mid T \in \mathcal{P}\}$ |
| sampled sets of cardinality 5 | and $F(R) \sim \text{Uniform}(-1, 1)$ for $R \in \mathcal{H}$ |
| and $F(T) \sim \text{Uniform}(-1, 1)$ for $T \in \mathcal{P}$ | |

## B.5   Shapley Value Approximation

We repeat the experiments of Section 4.3 under the metric of mean squared error relative to those computed by KernalSHAP under an inference budget of 10,000. In Figure 14, we find that PROXYSPEX uniformly outperforms KernelSHAP within this tested range. Just as with recall, with large enough $\alpha$, KernelSHAP eventually surpasses PROXYSPEX.

## B.6   Practical Implications

The practical implications of PROXYSPEX are significant, primarily revolving around its inference efficiency and the resulting speedups in generating faithful explanations for LLMs. A major challenge with existing interaction attribution methods, like SPEX, is the substantial number of model inferences required, which can be computationally expensive and time-consuming for large models. PROXYSPEX addresses this by leveraging a GBT proxy model, which dramatically reduces the number of inferences needed while maintaining or even improving explanation faithfulness.

Figure 15 presents the practical benefits in terms of wall clock time for achieving different levels of faithfulness ($R^2$) on the Sentiment Analysis (Figure 15a) and MS-COCO (Figure 15b) datasets. These plots clearly demonstrate the speedups achieved by PROXYSPEX. For example, in the sentiment analysis task using the smaller `DistilBERT` model, PROXYSPEX offers a speedup of approximately 3x, while for the larger `CLIP-ViT-B/32` model with MS-COCO, the speedup is around 5x when compared to methods that require more extensive sampling. This increased efficiency makes PROXYSPEX a more viable tool for interpreting complex LLMs in real-world scenarios where computational resources and time are often constrained.

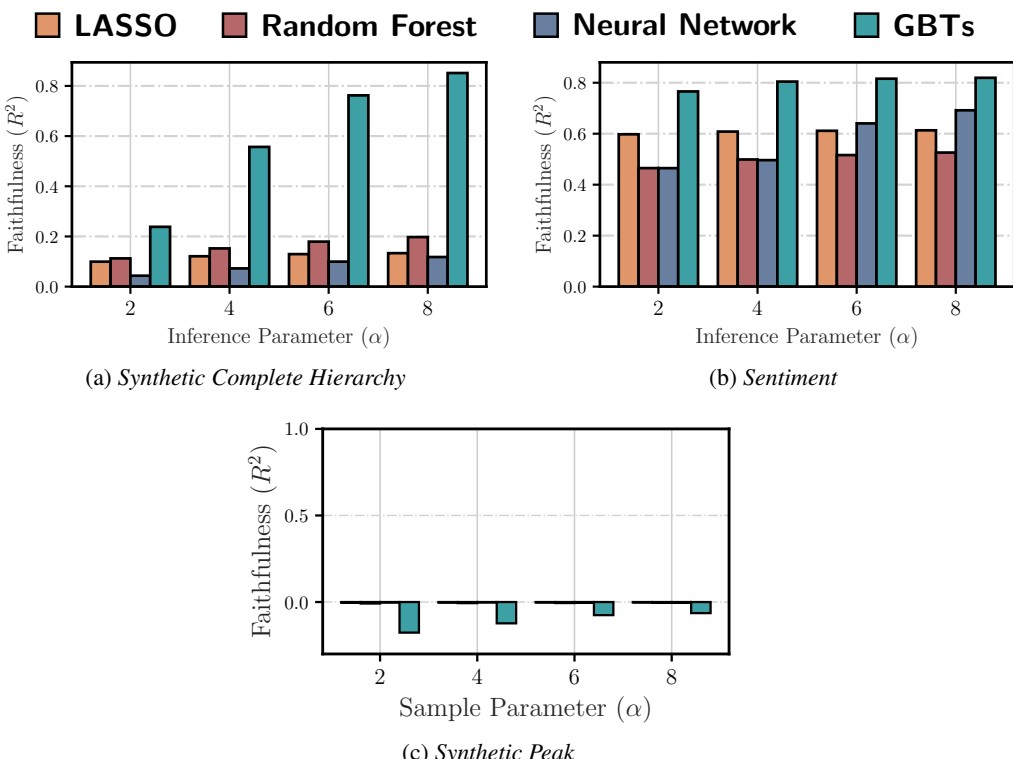

Figure 13: Comparison of proxy model faithfulness in capturing function structures. (a) Faithfulness of LASSO, Random Forest, Neural Network, and GBTs on a synthetic dataset with a complete hierarchical structure, across varying inference parameters ($\alpha$). (b) Faithfulness of the same proxy models on the Sentiment Analysis dataset across varying $\alpha$. (c) Faithfulness on a synthetic dataset with a sparse, non-hierarchical peak function, across varying $\alpha$, illustrating a limitation of GBTs for non-hierarchical structures.

## C  Case Study Details

### C.1  Data Attribution via Non-Linear Datamodels

The training masks and margin outputs were provided by [55], corresponding to their subsampling rate of 50% (i.e., half the training images were used to fit each model). See [55] for the hyperparameters selected. With $n = 50{,}000$ training samples, 300,000 training masks (model retrainings) were provided. This corresponds to $\alpha \approx 0.38$, which underscores the inference-efficiency of PROXYSPEX to identify strong interactions.

Utilizing these masks and margins, we randomly selected 60 test images (6 from each class) for analysis with PROXYSPEX. Below, in Figure 16 and Figure 17, we present the strongest second-order interactions of the first thirty of these selected test images. Figure 9 visualizes the six test images exhibiting the most significant third-order interactions identified through this analysis.

After fitting PROXYSPEX, we convert the Fourier interactions to Möbius using Appendix A.1. Since target and non-target images affect the test margin in opposite directions, we partition the interaction space into the following categories:

- *Target-class interactions $\mathcal{T}$:* Interactions composed exclusively of training images that share the same label as the held-out test image.

- *Non-target-class interactions $\mathcal{T}^c$:* Interactions where at least one training image in the set has a label different from that of the held-out test image.

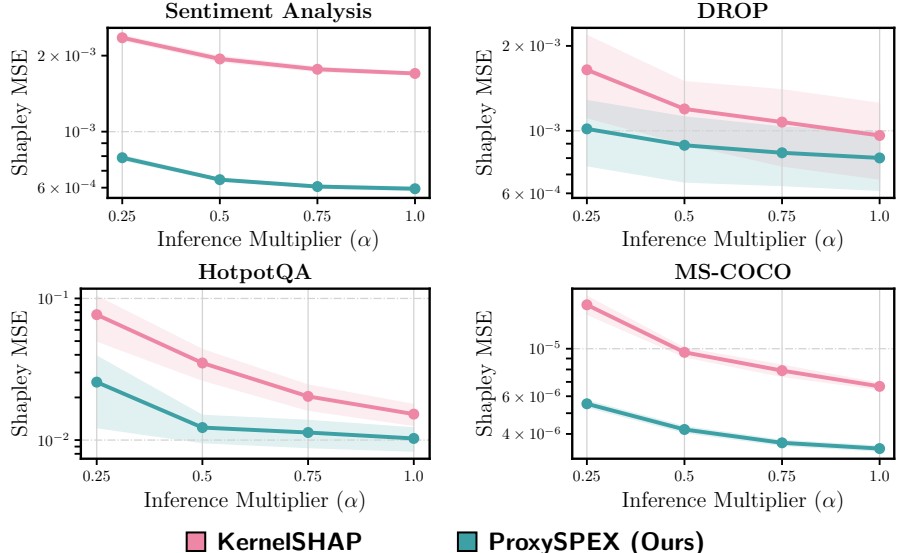

Figure 14: Shapley value mean square error after $\alpha \cdot n \log_2(n)$ inferences for multipliers $\alpha \in \{0.25, 0.5, 0.75, 1.0\}$. Across all four tasks and multipliers, within the tested range, PROXYSPEX provides a better approximation of the values computed under 10,000 inferences. Error bands indicate the standard deviation across ten different runs of the algorithms.

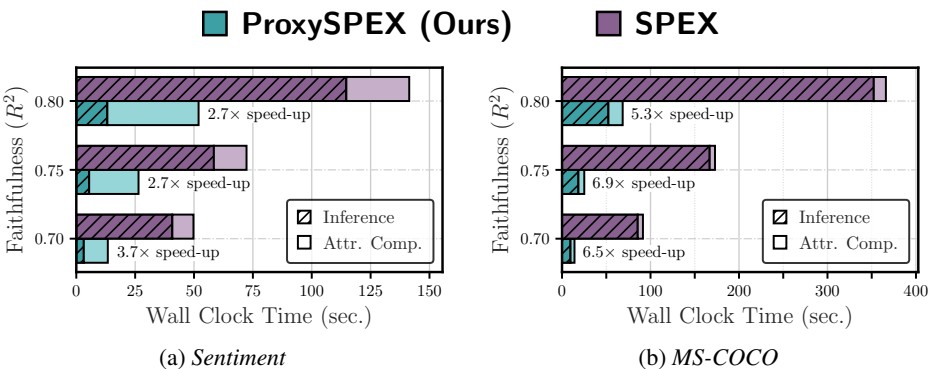

(a) *Sentiment*  (b) *MS-COCO*

Figure 15: Wall clock time demonstrating PROXYSPEX's efficiency. Comparison of wall clock time (seconds) required to achieve different levels of faithfulness ($R^2$) for PROXYSPEX, showing breakdown of inference time and attribution computation time. (a) Results on the Sentiment Analysis dataset with the `DistilBERT` model. (b) Results on the MS-COCO dataset with the `CLIP-ViT-B/32` model, highlighting speedups achieved by PROXYSPEX.

*Synergistic Interactions:* The top synergistic interaction $R^*$ of order-$r$ is defined as:

$$
\begin{aligned}
S^* &= \operatorname*{argmax}_{S \in \mathcal{T}, |S|=r} I^M(S) \\
T^* &= \operatorname*{argmin}_{T \in \mathcal{T}^c, |T|=r} I^M(T) \\
R^* &= \begin{cases} S^* & \text{if } |I^M(S^*)| \geq |I^M(T^*)| \\ T^* & \text{otherwise} \end{cases}
\end{aligned}
\tag{12}
$$

Visually, as presented in Figure 16 for $r = 2$, the interactions $R^*$ identified by this rule often involve training images that appear to work together to reinforce or clarify the classification of the held-out image, frequently by contributing complementary features or attributes. It is important to acknowledge that this definition serves as a heuristic and does not perfectly isolate synergy; For

example, the first frog image contains redundant bird images due to strong higher-order interactions involving these bird images.

*Redundant Interactions:* The top redundant interaction $R^*$ of order-$r$ is defined as:

$$
\begin{aligned}
S^* &= \underset{S \in \mathcal{T}, |S|=r}{\operatorname{argmin}} \ I^M(S) \\
T^* &= \underset{T \in \mathcal{T}^c, |T|=r}{\operatorname{argmax}} \ I^M(T) \\
R^* &= \begin{cases} S^* & \text{if } |I^M(S^*)| \geq |I^M(T^*)| \\ T^* & \text{otherwise} \end{cases}
\end{aligned}
\tag{13}
$$

Figure 17 demonstrates that this definition identifies redundant training images that are similar to the held-out image.

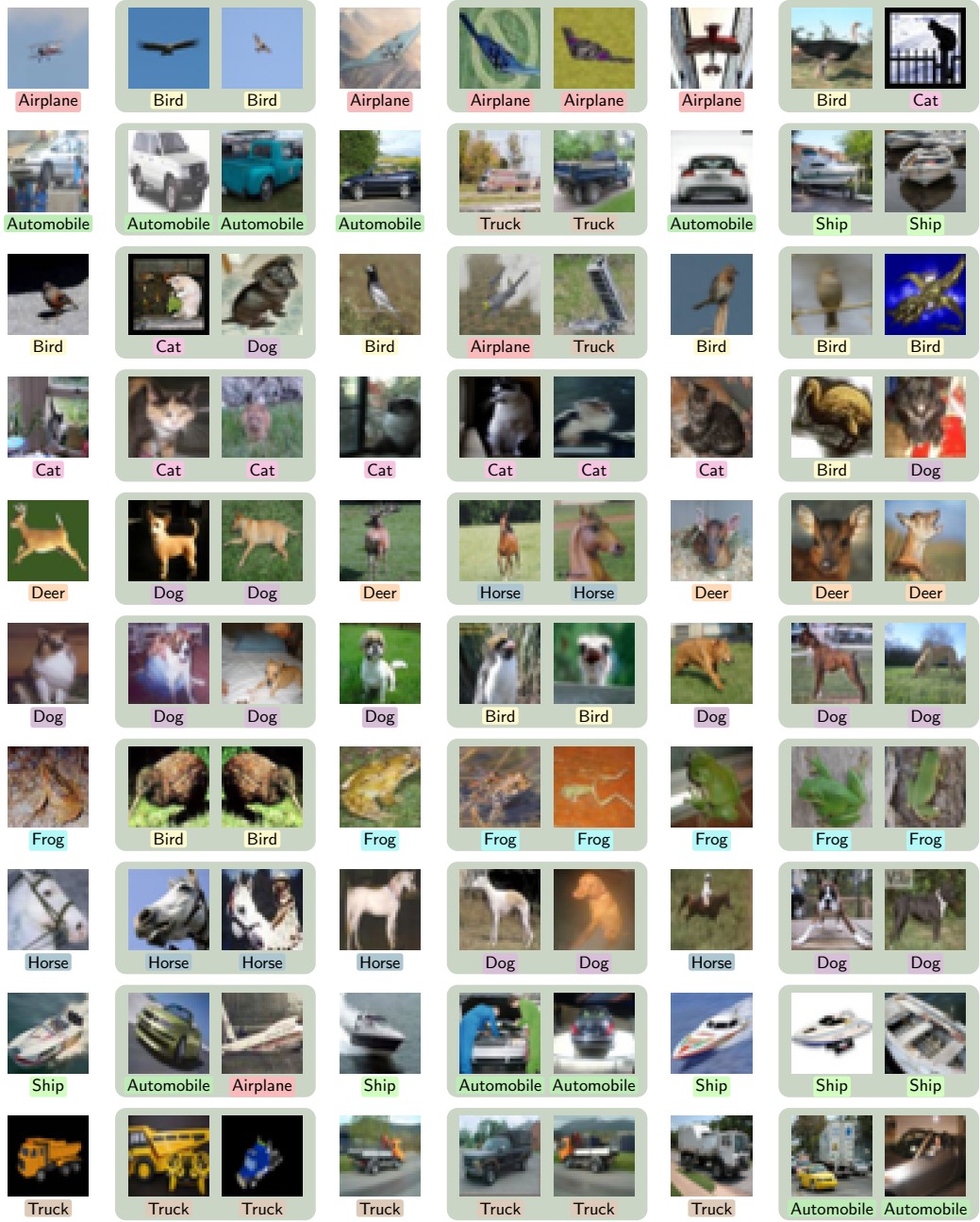

Figure 16: For 30 random held-out images, their corresponding top second-order *synergistic interaction* (green box).

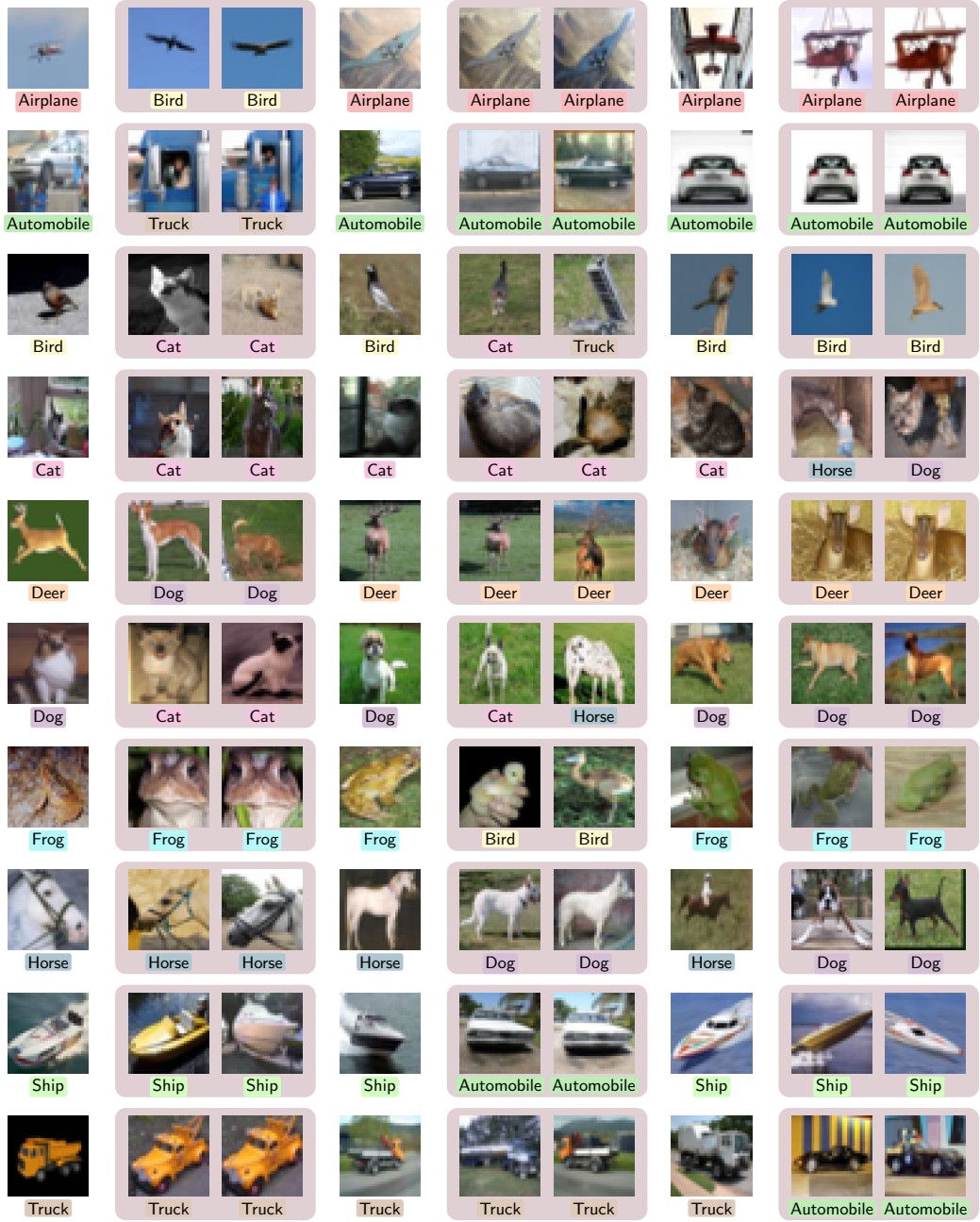

Figure 17: For 30 random held-out images, their corresponding top second-order *redundant interaction* (red box).

## C.2 Model Component Attribution

We study the influence of specific model components on task performance, using a controlled ablation methodology. Our experiments are conducted on `Llama-3.1-8B-Instruct` evaluated on the `high-school-us-history` subset of the MMLU dataset, a benchmark comprising multiple-choice questions.

MMLU includes 231 questions in the `high-school-us-history` subset. To perform pruning and then evaluate the ablated models, we split this data into two sets—training split $\mathcal{D}_{\text{train}}$ consisting of the first 120 questions and test split $\mathcal{D}_{\text{test}}$ with the remaining questions. We use accuracy as the evaluation metric, which is computed as the proportion of correctly answered multiple-choice questions on a given data split.

For an $L$ layer LLM, we let $[L]$ denote the set of layers and let $\mathcal{H}_\ell$ denote the set of attention heads in layer $\ell \in [L]$. For each experiment, we focus on a particular group of layers $\mathcal{L} \subseteq [L]$ within the model and denote the corresponding set of attention heads as $\mathcal{H}_\mathcal{L} = \bigcup_{\ell \in \mathcal{L}} \mathcal{H}_\ell$. The `Llama-3.1-8B-Instruct` model consists of $L = 32$ layers, each with 32 attention heads.

At each layer $\ell$ of the LLM, the output of the attention heads is combined into a latent representation by concatenating the outputs of the attention heads. Then, this latent vector is passed to the feed-forward network of layer $\ell$. To study the contribution of specific heads, we define an ablated model $\text{LLM}_S$ for any subset $S \subseteq \mathcal{H}_\mathcal{L}$. In $\text{LLM}_S$, the outputs of the heads in $\mathcal{H}_\mathcal{L} \setminus S$ are set to zero before the concatenation step. After concatenation, we apply a rescaling factor to the resulting latent vector at each layer $\ell \in \mathcal{L}$, equal to the inverse of the proportion of retained heads in that layer, i.e., $\frac{|\mathcal{H}_\ell|}{|S \cap \mathcal{H}_\ell|}$. This modified latent representation is then passed to the feed-forward network as usual.

We define $f_\mathcal{L}$ as

$$f_\mathcal{L}(S) \triangleq \text{Accuracy of LLM}_S \text{ on } \mathcal{D}_{\text{train}}, \tag{14}$$

and interpret $f_\mathcal{L}(S)$ as a proxy for the functional contribution of head subset $S$ to model performance, enabling quantitative analyses of attribution and interaction effects among attention heads.

**Pruning.** We perform pruning experiments across three different layer groups $\mathcal{L}$: initial layers ($\mathcal{L} = \{1, 2, 3\}$), middle layers ($\mathcal{L} = \{14, 15, 16\}$), and final layers ($\mathcal{L} = \{30, 31, 32\}$). Since each layer has 32 attention heads, we effectively perform ablation over $n = |\mathcal{H}_\mathcal{L}| = 96$ features (attention heads) in total. For a given group $\mathcal{L}$, we begin by estimating the function $f_\mathcal{L}$ using both LASSO and PROXYSPEX, based on evaluations of $f_\mathcal{L}(S)$ for 5000 subsets $S$ sampled uniformly at random. These estimates serve as surrogates for the true head importance function. We then maximize the estimated functions to identify the most important attention heads under varying sparsity constraints (target numbers of retained heads). We use the procedure detailed in Section 4.2 to identify heads to remove for both PROXYSPEX and LASSO. We also compare against a Best-of-$N$ baseline, in which the model is pruned by selecting the subset $S$ that achieves the highest value of $f_\mathcal{L}(S)$ among 5000 randomly sampled subsets at the target sparsity level.

**Evaluation.** In order to evaluate the performance of an ablated model $\text{LLM}_S$, we measure its accuracy on the test set using

$$g_\mathcal{L}(S) \triangleq \text{Accuracy of LLM}_S \text{ on } \mathcal{D}_{\text{test}}. \tag{15}$$

In Figure 10, we report the value of $g_\mathcal{L}(S)$ for the pruned models obtained by each method. We find that PROXYSPEX consistently outperforms both baselines, yielding higher test accuracy across all evaluated sparsity levels.

**Inference setup.** All experiments are run on a single NVIDIA H100 GPU, with batch size 50. Average runtime per ablation (i.e., evaluating $f_\mathcal{L}(S)$ once for a given $S$) is approximately 1.7 seconds. Therefore, collecting a training dataset $\{(S_i, f_\mathcal{L}(S_i))\}$ with 5000 training samples takes approximately 2.5 hours.

