# OpenReview forum: "ProxySPEX: Inference-Efficient Interpretability via Sparse Feature Interactions in LLMs"
_NeurIPS.cc/2025/Conference — NeurIPS 2025 spotlight_

### Official Review · Reviewer_ikfA · 2025-06-19

**Clarity:** 2
**Significance:** 3
**Originality:** 3
**Rating:** 5
**Confidence:** 3

**Summary:**

This work proposes ProxySPEX, a method for quantifying arbitrary-order interactions for many features. Specifically, the authors develop a method to represent and estimate *value functions* accurately, efficiently and sparsely. The approach stems from the empirical finding that interactions are often hierarchical (in the models considered), and thus may be well captured by gradient boosted trees. For a given input of interest, gradient boosted trees are trained to predict the value function (i.e. on different masked subsets of the input), after which a sparse set of Fourier coefficients are extracted. This approach to estimating and using the value functions is significantly more tractable than SPEX (a prior work also making use of the Fourier transform of the value function) achieving higher faithfulness with fewer inferences, and is shown to be superior to LASSO. The approach is shown to be effective for a number of model families and datasets, as well as in two case-studies.

**Questions:**

* It is reasonable to me that the Fourier coefficient $F(T)$ could be interpreted as the strength of the interaction in set $T$. However, as someone who does not have experience with Fourier Transforms on functions of the hypercube, it is not immediately clear to me why this is the case. It would be helpful to provide more detail as to why. As is, the paper states that $F(T)$ is the importance, and that “Eq. 1 represents an orthonormal transform onto a parity (XOR) basis”. Is this latter reason the explanation? If so, it could be expanded a bit.
* Complementary to the question above, is there a straightforward interpretation of the literal equation you provide for $F(T)$ as importance, i.e., aside from its interpretation as a Fourier coefficient? Maybe some sort of decomposition of the sum? It looks like $T$ benefits from having even intersections with $S$ that have large $f(S)$ and, when the intersection of $T$ and $S$ is odd, for $f(S)$ to be small, but this alone is not very intuitive to me as a measure of importance.

Including responses to one or both of the above questions in Section 3.1 would be valuable for improving the interpretability of the paper.

**Ethical Concerns:**

["NO or VERY MINOR ethics concerns only"]

**Final Justification:**

I maintain my positive rating of accept. This work addresses an important and challenging problem: tractably estimating interactions in high dimensions. The authors provide solid evidence that the hierarchical interactions motivating their approach can be found in the models they consider, a straightforward optimization (Eq 4) for using ProxySPEX to find influential feature groups, and a broad range of experimental results validating the efficacy of their method. The authors plan to fix the minor typos I have found as well as add citations and discussions for the relevant related works that I noted. The authors also provide a reasonable explanation of how the Fourier transform captures interaction importance. Finally, they address my issue regarding the difference between SPEX and ProxySPEX in their response to another reviewer, and I find that ProxySPEX still contains sufficient novelty.

**Limitations:**

Yes

**Quality:**

3

**Strengths And Weaknesses:**

**Strengths**
* The problem of identifying higher order feature interactions is both important and notoriously hard in high dimensions. Attempts to make this tractable for massive models, as in this work, are valuable.
* Section 3.2 and Figure 3 provide strong evidence that the models you consider indeed contain hierarchical interactions. Aside from providing motivation for your method, I believe that this is a potentially valuable finding for the community in general.
* The optimization described in Eq 4. Is a natural and efficient use of your $\hat{f}$ for finding influential groups of features.
* Your experiments clearly show ProxySPEX’s ability to achieve higher faithfulness with fewer inferences compared to SPEX and LASSO. Further, the experiments are thorough and wide-ranging covering multiple model families and datasets in the faithfulness and feature identification experiments, in addition to two case studies.

**Weaknesses**
* The method ProxySPEX is built upon, SPEX, is not made clear in this work. For example, I don’t think that the acronym SPEX is ever articulated. Aside from both using the Fourier transform it would be helpful to make the exact relationship between SPEX and ProxySPEX clear. From the naming and presentation of the work, my impression is that the major difference is the use of Gradient Boosted Trees, but more details would be helpful. I believe this can be achieved with some light modifications to the “Preliminary” section.

*Minor*
* There are 2 related works that I think the paper would benefit from including. In [1] the authors develop a pair-wise interaction quantification method which tractably extends to higher-order interactions, partially guided by the intuition that such interactions beget interactions of lower-order subsets, in their models. In [2] the authors create a method for extending a wide range of interpretability methods to directly quantify feature interactions – with an experimental focus on Shapley values and second-order interactions.
* (line 121) I believe $f$ should be $\hat{f}$.
* Eq 3. RHS: I believe this condition should be slightly rephrased as you are not summing over the coefficients but the sets which have the largest coefficients. Maybe something like “where $\mathcal{F}_k$ denotes the set of sets with the $k$ largest Fourier coefficients in $f$.” ?

[1] Tsang et al. How does this interaction affect me? interpretable attribution for feature interactions. Advances in Neural Information Processing Systems, 2020.

[2] Masoomi et al. Explanations of black-box models based on directional feature interactions. In International Conference on Learning Representations, 2021.

---

> ### Author Rebuttal · Authors · 2025-07-30
>
> 1. Thank you for your careful review as well as pointing out the relevant references [1] and [2]. Both should have been considered and we will add a discussion of both methods to the camera ready version.
>
> 2. Thanks, we will change $f$ to $\hat{f}$.
>
> 3.  We will replace this with the  more precise and correct statement. "where $\mathcal{F}_k$ denotes the set corresponding to the $k$ largest Fourier coefficients of $f$".
>
> Other Questions: Fourier transforms as interaction importance scores: One way to think about why a Fourier coefficient corresponds to an importance is to think of the Fourier transform $F$ as a decomposition of the function. For any given set $T$, $F(T) = 0$ implies that the function does not depend jointly on the effect of the features in $T$. On the other hand, the existence of a non-linear effect between features in $T$ (for example, a synergistic effect), implies a non-zero Fourier coefficient. The Fourier coefficients (scaled by a factor of $-2$) also precisely correspond to the Banzhaf Interaction Index from game theory [a], and thus inherits all interpretations.
>
> [a] Marichal, Jean-Luc, and Pierre Mathonet. "Weighted Banzhaf power and interaction indexes through weighted approximations of games." European journal of operational research 211.2 (2011): 352-358.

---

> > ### Comment · Reviewer_ikfA · 2025-08-01
> >
> > Thank you for your response. The authors address my recommendations in terms of presentation/notation and explain the interpretation of the Fourier transform as measuring interaction strength. I will maintain my already strong score of accept.

---

### Official Review · Reviewer_6aRb · 2025-06-30

**Clarity:** 2
**Significance:** 3
**Originality:** 3
**Rating:** 5
**Confidence:** 3

**Summary:**

This paper introduces a method for identifying important feature interactions. Given a function $f$ and input instance $x$, the method works by sampling a number of masks; evaluating $f$ on the masked instances of $x$; fitting a gradient boosted tree (GBT) to the dataset of mask/function output pairs; and extracting the Fourier representation of the GBTs. The empirical evaluation shows that the method outperforms a marginal feature attribution method, and outperforms another interaction attribution method using fewer function evaluations.

**Questions:**

- For # features removed (in figure 7): for ProxySPEX, is a feature a single word, or an interaction (a set of words)?
- Would it be possible to compare to one of the other interaction baselines on a dataset with fewer features?
- In sec. 5.1, would it be possible to give a more formal definition of redundant interactions and synergistic interactions? How are these identified?
- What is the relative faithfulness metric in figure 5? Is this relative to the GBT or to the underlying model?
- I did not find Figure 2 to be very helpful. Would it be possible to add more concrete example---for example, illustrating hierarchical features in an actual task?
- In line 164: "With $T$ trees of depth $d$ there are at most $O(T4^d)$ non-zero Fourier coefficients" Where does $O(T4^d)$ come from? For the decision trees in this paper, is $d = n$ (the number of features)?

**Ethical Concerns:**

["NO or VERY MINOR ethics concerns only"]

**Final Justification:**

This paper makes a clear methodological contribution by proposing a new method for finding feature interactions that empirically out-performs existing work, and scales more efficiently. In my original review, I listed a few main weaknesses. The authors have addressed these in the rebuttal.

1. Clarity about the method / relationship between SPEX and ProxySPEX. The authors have explained the differences in the rebuttal and plan to incorporate this clarification into the next version of the paper.

2. Empirical comparison to other methods. The authors have conducted additional evaluations comparing ProxySPEX to other methods for finding feature interactions.

Since the authors have addressed my main concerns, I am increasing my score to recommend that the paper is accepted.

**Limitations:**

yes

**Quality:**

3

**Strengths And Weaknesses:**

*Strengths*

- It is valuable to identify feature interactions, but computationally difficult. This method seems to offer an effective way to identify important interactions at lower cost (especially compared to the main prior work (SPEX).
- The method is evaluated on a variety of different settings. These include different feature granularities (words vs. sentenences), different modalities (text vs. images), and also two very different case studies (data attribution and model attribution). These evaluations indicate that the method has potential broad applicability.



*Weaknesses*

- Clarity: I found it difficult to understand the main methodological contributions from reading the paper. In particular, it was difficult to understand the difference in methodology between ProxySPEX and the main prior work (SPEX). The paper offers an extended discussion of the hierarchical nature of features in LLMs, but it was not so clear to me why GBTs leverage this hiearchy, and how this reduces the number of necessary function evaluations compared to prior work. I think the paper would be stronger if it gave a clearer description of the method, especially the differences between ProxySPEX and SPEX.
- The evaluations do not include comparison with other methods for identifying important interactions, with the exception of SPEX. The paper states that existing methods are too inefficient to be applied to the problem considered in the experiments, but I think it would be helpful to see comparisons on smaller size problem.
- Similarly, I had some difficulty understanding the empirical comparison between SPEX and ProxySPEX. Is the difference in performance due to the fact that SPEX requires more masks to perform well? How would the methods differ if you could enumerate all masks (e.g. on a toy problem)? Would ProxySPEX perform worse (given the GBT approximation)? Without a clearer description of the method, it is difficult to understand how exactly ProxySPEX improves over SPEX, and where it might be more limited.

*Overall:* This paper seems to offer a useful method that is substantially more efficient than methods from prior work, but clarity issues make it difficult to fully understand the method and the contribution relative to existing methods. I am open to increasing my score if the authors can address my comments above.

---

> ### Author Rebuttal · Authors · 2025-07-30
>
> Thank you for your review. We clarify some of the main points below, and hope to convince you that this work meets the criteria for acceptance.
>
> **Relationship between SPEX and ProxySPEX:** This was a common comment across many reviews. We will add a section carefully describing the relationship of the two approaches.
>
> Note that SPEX and ProxySPEX differ significantly. The only common factors in these two methods are that they:
>
> 1. Both exploit an empirically observed structure (sparsity and low-degree) for SPEX and (sparsity, low-degree and hierarchy) for ProxySPEX.
>
> 2. Both ultimately generate a Fourier transform to approximate the model.
>
> SPEX uses an algebraic structured sampling scheme, coupled with error correction decoding procedures to efficiently compute the interactions in the form of a Fourier transform. SPEX is a rather complex algorithm that involves many carefully designed components that work in concert to enable efficient estimation.
>
> ProxySPEX, in contrast, is much simpler. ProxySPEX uses random samples to learn a GBT proxy model. We then apply an efficient method for computing the fourier transform of the GBT model to extract the most important interactions from the proxy model.
>
> **Comparison to other interaction methods:** We stress that we believe the main advantage of ProxySPEX to be the scale at which it is able to operate. Due to this, we have primarily focus on comparing against other approaches that achieve scale, i.e., SPEX. As you suggested, we have conducted experiments that show that ProxySPEX also outperforms other approaches, even at smaller scales.
>
> Below, we repeat the faithfulness experiments for *Sentiment*, with 20 new movie reviews with $n\in [16,32]$ words and $n\in [64,128]$ words. As in Section 4.1, we vary the number of inferences by $\alpha \cdot n \log_2(n)$ and compute the faithfulness. In addition to SPEX and LASSO, we consider Faith-Banzhaf Interaction Indices [1] of up to $k$th order, which were shown in SPEX to achieve the best faithfulness among interaction attribution algorithms. Because this method enumerates all possible interactions of up to order $k$ (totaling $\approx n^k$ interactions), it becomes computationally intractable to consider >3rd order methods for the first experiment, and >2nd order methods for the second.
>
> > *Sentiment* for $n\in [16,32]$
> |           | $\alpha=2$ | $\alpha=4$ | $\alpha=6$ | $\alpha=8$ |
> |:---------:|:----------:|:----------:|:----------:|:----------:|
> | ProxySPEX |  **0.873** |  **0.919** |  **0.936** |  **0.946** |
> |    SPEX   |    0.510   |    0.621   |    0.750   |    0.801   |
> |   LASSO   |    0.727   |    0.744   |    0.746   |    0.749   |
> |  FBII 2nd |    0.846   |    0.891   |    0.905   |    0.911   |
> |  FBII 3rd |    0.836   |    0.898   |    0.920   |    0.932   |
>
>
> > *Sentiment* for $n\in [64,128]$
> |           | $\alpha=2$ | $\alpha=4$ | $\alpha=6$ | $\alpha=8$ |
> |:---------:|:----------:|:----------:|:----------:|:----------:|
> | ProxySPEX |  **0.805** |  **0.847** |  **0.868** |  **0.882** |
> |    SPEX   |    0.385   |    0.580   |    0.691   |    0.695   |
> |   LASSO   |    0.632   |    0.643   |    0.646   |    0.649   |
> |  FBII 2nd |    0.774   |    0.804   |    0.815   |    0.821   |
>
>
> **Hierarchical Nature of GBTs:**
> GBTs capture hierarchy in the feature space inherently through their structure. Each tree in the ensemble is, by its nature, a hierarchical model.
>
> Individual Trees: The first split in a tree, at the root, is on the feature that provides the most information gain  across the entire dataset. This is the most dominant feature in the hierarchy. As you move down the tree, further splits are made on other features within the subsets of data created by the previous split. For example, a split on feature B might only occur for data points where feature A was greater than a certain value.
>
> Boosting: The boosting process in GBTs refines this hierarchical modeling.
> Each new tree is trained to correct the errors of the existing ensemble. This means that later trees focus on capturing patterns and relationships that the earlier trees missed. If a complex hierarchical interaction between features was not fully modeled by the first few trees, subsequent trees will specifically target and model that remaining part of the relationship, further refining the hierarchy.
> For instance, an initial tree might capture the main effect of age. A later tree might then model a more subtle interaction, like how income has a different effect specifically for younger age groups, a hierarchical relationship that was a source of error for the earlier, simpler model.
>
> **Empirical Difference Between SPEX and ProxySPEX - number of masks**
> The primary empirical difference between SPEX and ProxySPEX is an order of magnitude decrease in the number of inference required to achieve the same faithfulness (see Figure 1). For large models (and LLMs in particular), due to the latency of inference, this results in a significant speed-up in wall clock time for computing interactions at scale.
> We would like to refer the reviewer to Figure 13 in appendix B.5 (The purple lines are for SPEX, and teal for ProxySPEX, the legend was ommitted in error, which will be corrected in the camera ready version). Here, SPEX requires much more time collecting data by performing inference as compared to ProxySPEX to achieve the same faithfulness. This is a result of ProxySPEX's ability to exploit hierarchical information to more efficiently identify interactions. SPEX, on the other hand does not directly exploit any relationship between interactions, only that the interactions themselves are sparse and low-degree.
>
> **Toy Problem - Mask Enumeration** The toy problem of mask enumeration is not an interesting one for comparing SPEX and ProxySPEX, since the most interesting mechanism is their ability to exploit structure to reduce the number of required masks. SPEX directly learns a Fourier transform, which is a universal approximator (since the transform is 1-to-1). In the case where it has full information about $f$ (access to all the masks), it has zero error. Similarly, in the ensemble limit and given sufficient depth, GBTs are universal approximators, and thus also achieve perfect reconstruction. Practically though, for some fixed ensemble size, there will be some non-zero approximation error, and SPEX would slightly outperform ProxySPEX in this setting.
>
> ## Questions:
>
> **Figure 7:** Here we are removing individual "features", which can be words or sentences, depending on the dataset. See the "datasets and models" part of section 4 for details on a per-dataset basis.
>
> **Section 5.1:** In the camera-ready version, we will discuss this in more detail. For your reference, precise definitions given in Appendix C.1. See rebuttal of review kopj for more details.
>
> **Figure 2:** With the additional page, we will improve this figure with a simple example from sentiment analysis in the camera-ready version.
>
> **Relative Faithfulness:** The relative faithfulness is relative to the ``full'' GBT model. That is, we are plotting how faithful the sparsified model is compared to the full GBT model. Our results show that sparsification does not hurt faithfulness while improving interpretability. The plots look similar when plotting absolute faithfulness (relative to the underlying function), but simply saturate at some fixed level (~90\%).  We will clarify this in the camera-ready version.
>
> **Line 164:** This result is provided in Gorji et al. [5] (top of page 6), for which we will provide a citation for this statement in the camera-ready version. This result comes from the fact that each leaf of a decision tree has at most $2^d$ non-zero Fourier coefficients. With each tree having at most $2^d$ leaves, and by linearity of tree ensembles and the Fourier transform over $T$ trees, there are at most $T\cdot 2^d \cdot 2^d = T4^d$ non-zero Fourier coefficients.
> For the decision trees in this paper, the depth is typically set to d=3 or d=5, selected through cross-validation. This is much fewer than $n$ number of features.
>
> [1] Tsai, C. P., Yeh, C. K., \& Ravikumar, P. (2023). Faith-shap: The faithful shapley interaction index. Journal of Machine Learning Research, 24(94), 1-42.

---

> > ### Comment · Reviewer_6aRb · 2025-08-05
> >
> > Thank you for the detailed response, and for clearing up some of my points of confusion about the differences between this method and SPEX. I appreciate the additional empirical results, and I appreciate the clarification. I think incorporating these changes will make the paper stronger, and I am updating my score accordingly.

---

### Official Review · Reviewer_EPSB · 2025-07-01

**Clarity:** 3
**Significance:** 4
**Originality:** 3
**Rating:** 5
**Confidence:** 3

**Summary:**

The paper focuses on the problem of "local approximation based explanations" which attempt to identify the features that were important for a particular prediction by approximating the model with a simpler/explainable function across some neighborhood.  Importantly, the paper considers interaction effects between these features.  Additionally, the paper shows that the same approach can be used for data attribution and model component attribution.


The paper proposes a method, ProxySPEX, which builds on an existing method, SPEX, based on two observations:
-  In addition to being sparse, the Fourier spectra of the interactions between features in LLMs is hierarchical
-  GBTs are very effective for learning this type of hierarchical structure

ProxySPEX then works by:
1.  Evaluating the model on a random sample of masked inputs
2.  Fitting the GBT to that data
3.  Efficiently computing the Fourier representation of the GBT using an existing method
4.  Refining those Fourier coefficients

The paper shows that this process produces more faithful explanations than SPEX and LASSO regression across a variety of datasets and number of masked samples used in Step 1.

Then the paper shows that different definitions of the value function can be used to enable ProxySPEX to be used for data attribution (identifying which training points where important to have for the model's prediction at this point and which were redundant) and model component attribution (identifying which attention heads can be pruned).

**Questions:**

Could the authors elaborate on Step 4?  I'm still not clear on exactly how this works.

While reading the paper, I was confused about what the "local neighborhood" ProxySPEX was approximating the model over.  I eventually realized that this was "the set of all randomly masked inputs".  But it would be nice if this was made clearer earlier.

It'd be great if the authors commented on why SHAP isn't expected to perform well in this setting.

**Ethical Concerns:**

["NO or VERY MINOR ethics concerns only"]

**Final Justification:**

The authors' responses addressed my concerns.

Looking at the other reviews, I think the main concern is with the novelty of ProxySPEX.  While many of the "pieces" required to build ProxySPEX already existed, I think there is value in putting them together.  Especially when the results are good.  As a result, I don't think that the concerns over novelty are sufficient to reject the paper.

Overall, I still recommend that the paper is accepted.

**Limitations:**

Yes

**Quality:**

4

**Strengths And Weaknesses:**

Strengths:
-  Quality:  The paper supports its claims with well designed and extensive experiments
-  Clarity:  Most of the paper is clearly written.  The next section has some questions/suggestions regarding this.
-  Significance:  The problem studied is an important one and the proposed method is a clear improvement along multiple axes (computational efficiency and efficacy).
-  Originality:  I'm not deeply familiar with the related work, but the two observations that justify ProxySPEX were interesting to read and think about.

Weaknesses:  Clarity on SHAP vs LIME (and other related methods)
-  It is very common for papers to treat SHAP (and related methods) and LIME (and related methods) as interchangeable because they are both "local explanations that assign 'importance' to features".
-  However, they are better thought of as separate types of explanations.  SHAP is an example of a "local feature attribution" explanation, which identifies "important" features (usually by measuring how the model's predictions change when a feature is "removed").  LIME (and ProxySPEX) are "local approximation based" explanations which explicitly try to approximate the model across some "neighborhood".
-  The fact that the "neighborhood" in this paper is the "the set of all randomly masked inputs" blurs the line between these types of explanations, but the evaluation metric (faithfulness) is clearly measuring approximation accuracy accuracy across it.  Given that, it is entirely expected for SHAP (or anything other local feature attribution explanation) to perform poorly on it (they are optimized for something else).  Especially when compared to methods that explicitly optimize for it.
-  This detail hardly matters for this paper.  But it would be nice if papers treated these method categories with more care because it would go a long way towards reducing the confusion about when and why one method works better than another (and also towards reducing the amount of non-sensical comparisons run or requested by reviewers).

---

> ### Author Rebuttal · Authors · 2025-07-30
>
> Dear reviewer, thank you for your comments and review. We appreciate your effort and insightful follow-up questions.
>
> **Weakness Section:** This is a good point, and is certainly worthy of a more nuanced follow-up discussion in the camera ready version. We should expect Shapley Values, which are not optimized for faithfulness, to perform poorly in this task, which is observed in Appendix B.5.1 of [1].
>
> The structure of the Shapley Kernel is more suited to the feature identification experiments in Section 4.2 of this paper (where we try to identify the top $r$ features). Here too, however, we find that Shapely values perform comparably to LASSO/LIME which we significantly outperform. Similar results are observed in Section 6.3 of [1]. Interestingly we also find that using the learned Fourier transform of ProxySPEX to estimate the Shapley Values often generates a better estimate (in terms of MSE, correlation, etc.) of the Shapley Values than standard approaches such as KernelSHAP.
>
> **Regression:**  We will extend our discussion of the use of a final regression step. Empirically, we observe that in most cases, there is a small positive effect on the final faithfulness, typically most impactful in very high-noise settings.
>
> **Follow-up discussion:**
>
> *Some discussion of "approximation" vs. "importance"*:
> LIME is typically viewed as an "approximation" while Shapley Values are not. It should be noted though, the distinction between "approximation" and "importance" is somewhat arbitrary. Ultimately, most attribution scores can be defined as the solution to an approximation problem with a suitably defined distance metric (loss) and optionally some regularization. For SPEX and ProxySPEX, (and often LIME) that is the $\ell_2$ distance metric, while for the Shapley Value that distance metric is the Shapley Kernel (this is the essence of Theorem 2 in Lundberg et al. 2017).
>
> The real distinction seems to be that local "approximation" based approaches are typically evaluated in terms of their approximation loss, while Shapley Values, are typically not. In the camera ready version, we will add some discussion of this, and highlight our definition of "distance", which is, as you described, the set of all randomly masked inputs.
>
> Interestingly, and complementary to the above discussion, one can  use the ProxySPEX approximation to compute Shapley values using the closed-form conversion in Appendix A.1. Evidence suggests that this is efficient strategy. Using the *Sentiment* dataset and model, we first approximated the Shapley values by running KernelSHAP (Lundberg et al. 2017) under 100,000 model inferences, treating these as the ground truth value. We then compared the approximation quality of the Shapley values computed via KernelSHAP and ProxySPEX under just 100 and 1,000 model inferences, measuring the mean-squared error, spearman rank correlation, and the recall of the top 10 Shapley values. Results are averaged over the 20 movie reviews in the dataset.
>
> |            |  MSE (100) ↓ | MSE (1000) ↓ | Spear. Corr. (100) ↑ | Spear. Corr. (1000) ↑ | Recall@10 (100) ↑ | Recall@10 (1000) ↑ |
> |:----------:|:----------:|:----------:|:------------------:|:-------------------:|:---------------:|:----------------:|
> |  ProxySPEX | **0.0015** | **0.0004** |     **0.2197**     |      **0.6168**     |    **0.300**    |     **0.725**    |
> | KernelSHAP |   0.0728   |   0.0071   |       0.0947       |        0.3167       |      0.085      |       0.400      |
>
> The approximation quality is better under all three metrics in this regime. A more thorough comparison with other approximation algorithms is  needed, but it seems that leveraging Fourier structural properties (sparsity, hierarchy and low-degree structures) is valuable, not just in an "approximation" sense, but also in determining Shapley Value-like notions of "importance" when model evaluation is very costly.
>
> [1] Kang, J. S., Butler, L., Agarwal, A., Erginbas, Y. E., Pedarsani, R., Yu, B., Ramchandran, K. SPEX: Scaling Feature Interaction Explanations for LLMs. In Forty-second International Conference on Machine Learning.

---

> > ### Comment · Reviewer_EPSB · 2025-08-05
> > **Response to rebuttal**
> >
> > Once these discussions are incorporated into the paper (or at least referenced in the main body and then expanded on in the appendix), my questions will have been addressed.  Thanks!

---

### Official Review · Reviewer_kopj · 2025-07-03

**Clarity:** 3
**Significance:** 3
**Originality:** 3
**Rating:** 5
**Confidence:** 2

**Summary:**

In this paper, the authors develop a method that enhances the interpretability of LLMs. In particular, they want to identify interactions responsible for the prediction in a downstream task for a given model $f$ and instance $x$. To that end, building on the SPEX framework, the authors study the value function, which is the output of the LLMs $f$ with some tokens randomly replaced by the [MASK] token and approximates this function using the Fourier Transform.

As opposed to SPEX, which frames the problem of finding the optimal Fourier coefficient as a decoding problem, ProxySpex proposes to train auxiliary gradient boosting trees on a dataset of inference values and associated masks. The intuition (backed by experimental results) is that the LLMs learns hierarchical features, i.e. features part of an interaction group are more likely to be selected as singletons as well (or being part of a sub-group). Leveraging recent results on boosting trees shows that they also capture such interaction and allow for the fast computation of the Fourier Coefficient with them. The resulting coefficients are further filtered to allow for even more sparsity.

This approximation is constructed for 4 different datasets with different downstream tasks and is compared to other approximations obtained with LASSO and the original SPEX. It is shown that the interactions recovered by Proxy Spex are more influential on the performance than its two counterparts and the approximation is more faithful.  Proxy Spex is further applied to different use cases beyond feature attribution ("datamodelling" as [1] and head pruning [1]) and shows promising results on these test cases.

**Questions:**

- Have the authors explored other mask sampling methods to allow a better evaluation of the model ? For instance, using the attention maps in the mask design directly could lead to the combination or subgroups of features?
- I am a bit unsure of the novelty of some parts of the paper. For instance, are the definitions of redundant and synergistic interaction new?
- The hierarchy discovered in the model seems similar to this prior training, which suggests that one needs to go through unigram, then bigram ... to discover the results [1]. It could be interesting to discuss these results in the paper.

[1] Edelman, E., Tsilivis, N., Edelman, B., Malach, E., & Goel, S. (2024). The evolution of statistical induction heads: In-context learning markov chains. Advances in neural information processing systems, 37, 64273-64311.

**Ethical Concerns:**

["NO or VERY MINOR ethics concerns only"]

**Final Justification:**

The answer from the authors addressed my concerns regarding the novelty, and I have raised my score accordingly.

**Limitations:**

The limitations of the models are correctly addressed. However, I don't think I see the error bars mentioned for the statistical significance of the results.

**Quality:**

4

**Strengths And Weaknesses:**

- **Quality and Originality** :
	- (+) Clear motivation : All the intuitions (hierarchical nature of the interaction of the LLM or the GBT, sparsity of the resulting approximation) used to motivate the model is experimentally investigated.
	- (+) Extensive experiments conducted on four diverse datasets (NLP and vision tasks) show ProxySPEX's superior faithfulness, efficiency, and interpretability compared to SPEX and LASSO.
	- (-)Due to the closeness with existing methods (mostly SPEX), I think a proper discussion to properly distinguish this method from the other is required.
	- (-) The regression-based refinement step is only briefly discussed, but might be crucial in practice for stability and accuracy.

- **Clarity**:  (+) The paper is very clear and the experiments are carefully described.

- **Significance**:  The method develop in this paper is very close to existing methods (SPEX) and ideas from other papers in the field (approximating the value function with Fourier transforms, leveraging GBT for Fourier Coefficients...), but the experimental results are impressive and constitute a nice advancement in the field. The extension to the two case studies on data attribution and attention head pruning demonstrates ProxySPEX's applicability beyond standard feature attribution and large potential to other applications. Different metrics and concepts (DSR, Synergetic interactions) were introduced and should help ML practitioners better understand both datasets and models.

---

> ### Author Rebuttal · Authors · 2025-07-30
>
> Thank you for your careful review. We will discuss and clarify some of the main points below, and hope to convince you that this is not a borderline case.
>
> **Relationship between SPEX and ProxySPEX:** This was a common comment across many reviews. We will add a section carefully describing the relationship of the two approaches.
>
> Note that SPEX and ProxySPEX differ significantly. The only common factors in these two methods are that they:
>
> 1. Both exploit an empirically observed structure (sparsity and low-degree) for SPEX and (sparsity, low-degree and hierarchy) for ProxySPEX.
>
> 2. Both ultimately generate a Fourier transform to approximate the model.
>
> SPEX uses an algebraic structured sampling scheme, coupled with error correction decoding procedures to efficiently compute the interactions in the form of a Fourier transform. SPEX is a rather complex algorithm that involves many carefully designed components that work in concert to enable efficient estimation.
>
> ProxySPEX, in contrast, is much simpler. ProxySPEX uses random samples to learn a GBT proxy model. We then apply an efficient method for computing the Fourier transform of the GBT model to extract the most important interactions from the proxy model.
>
> **Regression:** We will extend our discussion of the use of a final regression step. Empirically, we observe that in most cases, there is a small positive effect on the final faithfulness, but is not a critical aspect of ProxySPEX.
>
> **Other sampling methods:** We would like to emphasize that ProxySPEX is meant to be a *model agnostic* approach to identifying sparse features interactions. That being said, it is certainly an interesting research direction to try to extract an even more powerful prior beyond sparsity and hierarchy by looking at internal components (that are computed anyways) during the process of inference. During the process of developing ProxySPEX, we did study correlations between our extracted interactions and the one suggested by attention heads, but more work would be required to have a complete and operationally useful understanding of this relationship.
>
> **Definitions of Synergy and Redundancy:** Creating definitions of redundancy and synergy in interactions for classification was a problem we faced when developing this work. As far as we are aware, there do not exist standard definitions for these, as it is typically considered computationally intractable to compute even pairwise interactions between training data. Our definitions come from the notion of *Harsanyi dividends* in game theory, and how they relate to subadditive and superadditive game structures [1]. Importantly, Harsanyi dividends are equivalent to the Mobius coefficients, which we can compute due to efficient transformations between the Fourier and the Mobius transform. In the camera ready version, we will more prominently discuss these definitions.
>
> **Edelman et al.** This paper has some fascinating observations. It seems likely that the models trained in Edelman et al. would exhibit a hierarchical interaction structure, just like the larger, production-size models we considered in this paper. We would certainly discuss this connection in the final revision of our manuscript.
>
> [1] Dehez, P. (2017). On Harsanyi dividends and asymmetric values. International Game Theory Review, 19(03), 1750012.

---

> > ### Comment · Reviewer_kopj · 2025-08-06
> >
> > Thank you for your detailed response, which successfully addressed my concerns. The clarification regarding the relationship between SPEX and ProxySPEX was particularly helpful, while the confirmation of the originality of the study on the different types of interaction strengthened the novelty of this work.
> >
> > Assuming these discussions are incorporated into the final version of the paper, I would be glad to recommend this paper for publication. I have increased my score accordingly.

---

### Decision · Program_Chairs · 2025-09-17

**Decision:**

Accept (spotlight)

**Comment:**

This paper proposes ProxySPEX, an inference-efficient approach for recovering influential higher-order feature interactions around a test input by fitting a gradient-boosted tree proxy on masked inputs and then extracting a sparse Fourier representation from the proxy. The key empirical premise is that local interaction spectra in the studied models are not only sparse and low-degree but also hierarchical, in the sense that strong higher-order interactions tend to co-occur with their subsets. Across sentiment classification with DistilBERT, question answering with Llama-3/3.1, and CLIP image-text matching, the method achieves higher faithfulness than LASSO and approaches or exceeds SPEX with roughly an order of magnitude fewer model queries. The paper also presents case studies on data attribution and attention-head interactions that show the surrogate can support downstream interventions such as feature removal and head pruning.

The strongest aspects are the clear computational win relative to prior interaction attribution at comparable faithfulness, the breadth of evaluation across modalities and value functions, and a concrete pipeline that produces compact, interpretable spectra that can be reused for analysis beyond attribution. The empirical characterization of hierarchical interactions is useful in its own right and helps justify the proxy choice. Reviewers found the experiments well designed overall, and the added results during rebuttal strengthened the case for practicality.

The main weaknesses raised during review were about novelty relative to SPEX, clarity of the relationship between the two methods, and the scope of comparisons. Several reviewers also asked for clearer exposition of the masking neighborhood and the regression refinement step, and for stronger context on why SHAP-style methods are not competitive under the paper’s faithfulness metric. My own reading also flags an external-validity question as the approach leans on the hierarchy prior, and performance as this prior weakens is not fully mapped. Finally, the paper emphasizes reduced inference counts however lacks a fully budgeted comparison that includes proxy fitting time.

The rebuttal and discussion were productive. Reviewer kopj initially questioned novelty and asked about the regression step and the authors provided a precise delineation of ProxySPEX versus SPEX and committed to adding a comparative section to the paper. They also clarified that regression yields small but consistent gains without being critical. Reviewer EPSB asked for a clearer statement of the neighborhood and why SHAP should not be expected to perform well under the chosen metric and the authors addressed both points and also showed that ProxySPEX’s Fourier surrogate can efficiently approximate Shapley values. Reviewer 6aRb’s initial concerns centered on method clarity and comparisons to which the authors supplied additional small-scale experiments against interaction baselines and a more explicit account of how GBTs leverage hierarchy and why this reduces query counts relative to SPEX. Finally, reviewer ikfA requested clearer connections to related work on interactions and minor notation fixes and the authors agreed to incorporate the citations. They also clarified the interpretation of Fourier coefficients as interaction strengths. After rebuttal, there is consensus amongst the reviwers (and myself) for clear acceptance of the paper as the earlier concerns about clarity and novelty have been addressed to satisfaction.